# Toward Optimal Policy Population Growth in Two-Player Zero-Sum Games

Stephen McAleer[*1], JB Lanier[2], Kevin A. Wang[2], Pierre Baldi[2], Tuomas Sandholm[1] and Roy Fox[2]

[1]Department of Computer Science, Carnegie Mellon University
[2]Department of Computer Science, University of California, Irvine
[*]Corresponding author: smcaleer@cs.cmu.edu

## Abstract

In competitive two-agent environments, deep reinforcement learning (RL) methods like *Policy Space Response Oracles (PSRO)* often increase exploitability between iterations, which is problematic when training in large games. To address this issue, we introduce *anytime double oracle (ADO)*, an algorithm that ensures exploitability does not increase between iterations, and its approximate extensive-form version, *anytime PSRO (APSRO)*. ADO converges to a Nash equilibrium while iteratively reducing exploitability. However, convergence in these algorithms may require adding all of a game's deterministic policies. To improve this, we propose *Self-Play PSRO (SP-PSRO)*, which incorporates an approximately optimal stochastic policy into the population in each iteration. APSRO and SP-PSRO demonstrate lower exploitability and near-monotonic exploitability reduction in games like Leduc poker and Liar's Dice. Empirically, SP-PSRO often converges much faster than APSRO and PSRO, requiring only a few iterations in many games.

## 1 Introduction

In competitive two-agent environments, also known as zero-sum games, deep reinforcement learning (RL) methods based on the *double oracle (DO)* algorithm (McMahan et al., 2003), such as *Policy Space Response Oracles (PSRO)* (Lanctot et al., 2017), are some of the most promising methods for finding approximate Nash equilibria in large games. One reason is that such methods are simple to use with existing RL methods and naturally provide a measure of approximate exploitability. The exploitability of a policy is defined as the performance against a worst-case opponent, and it is optimal at zero when the policy is a Nash equilibrium. A second reason is that these methods effectively prune the game tree by only considering mixtures over policies that are already trained to be effective best responses. Finally, they can be used in games with large or continuous action spaces because they do not require full game-tree traversals. Methods based on PSRO such as AlphaStar (Vinyals et al., 2019) and Pipeline PSRO (McAleer et al., 2020) have achieved state-of-the-art performance on Starcraft and Barrage Stratego, respectively.

PSRO-based methods iteratively add RL best-response policies to a population. The best response for each player trains against a restricted distribution over the opponent's existing population of policies. To find this restricted distribution, a Nash equilibrium (a pair of mutually best-responding policies) is computed in a restricted single-step game where each action corresponds to choosing a policy from the population. As PSRO iterations progress, an optimal distribution over these population policies will approximate a Nash equilibrium in the full game.

In practice, however, PSRO is terminated early in large games. This can be a problem because the PSRO restricted distribution over the population policies is not guaranteed to decrease in exploitability every iteration. As a result, if PSRO is terminated early, the final restricted distribution could potentially be arbitrarily more exploitable than the initial one.

In this paper, we first propose a new double oracle variant, *anytime double oracle (ADO)* that, in each iteration, finds the least-exploitable restricted distribution over the population policies of each player. This algorithm is called *anytime*, in the sense that it can be stopped in any iteration and return a

solution that is not worse than the previous iteration. We then present an approximate extensive-form RL version called *anytime PSRO (APSRO)*.

*Anytime double oracle (ADO)* can be viewed as a modification of the *range of skill (ROS)* algorithm (Zinkevich et al., 2007) that finds a restricted Nash equilibrium over two restricted games, one per player. Each player's restricted game is defined such that their strategies are restricted to be within their population, but the opponent is unrestricted. For each player, ADO adds to the opponent's population a best response to the player's NE restricted distribution. ADO is guaranteed not to increase exploitability from one iteration to the next, while also being guaranteed to converge to a Nash equilibrium in a number of iterations at most equal to the number of pure strategies in the game.

*Anytime policy-space response oracles (APSRO)* updates the restricted distribution using a no-regret algorithm trained against a single approximate best response from the opponent. This opponent approximate best response is itself being continually trained via reinforcement learning against the restricted distribution. We find empirically that APSRO tends not to increase exploitability as much as PSRO and can greatly outperform PSRO in some domains.

However, because common implementations of PSRO add pure-strategy (i.e. deterministic) best responses in each iteration, PSRO may still need to add many policies to the population before they can support a Nash equilibrium. In fact, in certain games, all pure strategies will be added before finding a Nash equilibrium. This is because many games require mixing over a large number of pure strategies to arrive at a Nash equilibrium. Furthermore, before termination, the restricted distribution over population policies can be arbitrarily exploitable, even if it decreases monotonically until then.

In addition to introducing APSRO, we also build on APSRO by adding to the population in each iteration a stochastic policy that is trained via an off-policy procedure. A key insight is that mixed strategies (i.e. stochastic policies) can lower the exploitability of a population more than pure strategies. To see this, note that a Nash equilibrium strategy is an optimal strategy to add because the least-exploitable distribution over the resulting population will also be a Nash equilibrium strategy. If all Nash equilibria are mixed, as is often the case, then no pure strategy can be added to the population that reduces exploitability as much as the mixed strategy Nash equilibrium.

Although finding the optimal strategy to add is as hard as solving the original game, we find that adding a rough approximation to the optimal strategy can offer striking empirical benefits in quickly reducing the exploitability of the restricted distribution. We present Self-Play PSRO (SP-PSRO), which, similarly to APSRO, learns a restricted distribution over the population via no regret against the opponent's best response. Additionally, SP-PSRO trains off-policy a new strategy against the opponent's best response. At the end of each iteration, SP-PSRO adds two strategies to the population: (1) the time-average of this new strategy and (2) the best response to the opponent's restricted distribution. Section 5 clarifies this algorithm using formal notation. In large games like dark chess and Starcraft, where PSRO may never converge, the early performance holds paramount importance. Our approach with SP-PSRO is tailored to this reality, ensuring robust performance from the outset. Recognizing that the completion of the full training procedure in such extensive games is a rare occurrence, the anytime property of our proposed method takes on a critical role, delivering viable strategies at any stage of the iterative process.

By training the new strategy off-policy, SP-PSRO requires the same amount of experience in each iteration as APSRO and PSRO. Experiments on normal-form games and extensive-form games such as Liar's Dice, Battleship, and Leduc Poker suggest that SP-PSRO can learn policies that are dramatically less exploitable than APSRO and PSRO. Our empirical results demonstrate SP-PSRO's superior performance in reducing exploitability before convergence across various games, a testament to its practical effectiveness. While APSRO serves as a foundational concept in our research, the leap to SP-PSRO marks a significant advancement, particularly in terms of reducing exploitability before PSRO has neared convergence.

To summarize, our contributions are as follows:

- We introduce a version of double oracle that does not increase in exploitability, called anytime double oracle (ADO) and its extensive-form approximation, anytime PSRO (APSRO).

- We present an enhancement to APSRO, termed Self-Play PSRO (SP-PSRO). In each iteration, without requiring extra environment steps, it incorporates an additional mixed strategy aimed at reducing our population's exploitability.

## 2 BACKGROUND

We consider extensive-form games with perfect recall (Hansen et al., 2004). An extensive-form game progresses through a sequence of player actions and has a *world state* $w \in \mathcal{W}$ at each step. In an $N$-player game, $\mathcal{A} = \mathcal{A}_1 \times \cdots \times \mathcal{A}_N$ is the space of joint actions for the players. $\mathcal{A}_i(w) \subseteq \mathcal{A}_i$ denotes the set of legal actions for player $i \in \mathcal{N} = \{1, \ldots, N\}$ at world state $w$ and $a = (a_1, \ldots, a_N) \in \mathcal{A}$ denotes a joint action. At each world state, after the players choose a joint action, a transition function $\mathcal{T}(w, a) \in \Delta^{\mathcal{W}}$ determines the probability distribution of the next world state $w'$. Upon transition from world state $w$ to $w'$ via joint action $a$, player $i$ makes an *observation* $o_i = \mathcal{O}_i(w, a, w')$. In each world state $w$, player $i$ receives a reward $\mathcal{R}_i(w)$. The game ends when the players reach a terminal world state. In this paper, we consider games that are guaranteed to end in a finite number of actions.

A *history* is a sequence of actions and world states, denoted $h = (w^0, a^0, w^1, a^1, \ldots, w^t)$, where $w^0$ is the known initial world state of the game. $\mathcal{R}_i(h)$ and $\mathcal{A}_i(h)$ are, respectively, the reward and set of legal actions for player $i$ in the last world state of a history $h$. An *information set* for player $i$, denoted by $s_i$, is a sequence of that player's observations and actions up until that time $s_i(h) = (a_i^0, o_i^1, a_i^1, \ldots, o_i^t)$. Define the set of all information sets for player $i$ to be $\mathcal{I}_i$. The set of histories that correspond to an information set $s_i$ is denoted $\mathcal{H}(s_i) = \{h : s_i(h) = s_i\}$, and it is assumed that they all share the same set of legal actions $\mathcal{A}_i(s_i(h)) = \mathcal{A}_i(h)$.

A player's *strategy* $\pi_i$ is a function mapping from an information set to a probability distribution over actions. A *strategy profile* $\pi$ is a tuple $(\pi_1, \ldots, \pi_N)$. All players other than $i$ are denoted $-i$, and their strategies are jointly denoted $\pi_{-i}$. A strategy for a history $h$ is denoted $\pi_i(h) = \pi_i(s_i(h))$ and $\pi(h)$ is the corresponding strategy profile. When a strategy $\pi_i$ is learned through RL, we refer to the learned strategy as a *policy*.

The *expected value (EV)* $v_i^\pi(h)$ for player $i$ is the expected sum of future rewards for player $i$ in history $h$, when all players play strategy profile $\pi$. The EV for an information set $s_i$ is denoted $v_i^\pi(s_i)$ and the EV for the entire game is denoted $v_i(\pi)$. A *two-player zero-sum* game has $v_1(\pi) + v_2(\pi) = 0$ for all strategy profiles $\pi$. The EV for an action in an information set is denoted $v_i^\pi(s_i, a_i)$. A *Nash equilibrium (NE)* is a strategy profile such that, if all players played their NE strategy, no player could achieve higher EV by deviating from it. Formally, $\pi^*$ is a NE if $v_i(\pi^*) = \max_{\pi_i} v_i(\pi_i, \pi_{-i}^*)$ for each player $i$.

The *exploitability* $e(\pi)$ of a strategy profile $\pi$ is defined as $e(\pi) = \sum_{i \in \mathcal{N}} \max_{\pi_i'} v_i(\pi_i', \pi_{-i})$. A *best response (BR)* strategy $\mathbb{BR}_i(\pi_{-i})$ for player $i$ to a strategy $\pi_{-i}$ is a strategy that maximally exploits $\pi_{-i}$: $\mathbb{BR}_i(\pi_{-i}) = \arg\max_{\pi_i} v_i(\pi_i, \pi_{-i})$. An $\epsilon$-*best response* ($\epsilon$-BR) strategy $\mathbb{BR}_i^\epsilon(\pi_{-i})$ for player $i$ to a strategy $\pi_{-i}$ is a strategy that is at most $\epsilon$ worse for player $i$ than the best response: $v_i(\mathbb{BR}_i^\epsilon(\pi_{-i}), \pi_{-i}) \geq v_i(\mathbb{BR}_i(\pi_{-i}), \pi_{-i}) - \epsilon$. An $\epsilon$-*Nash equilibrium* ($\epsilon$-NE) is a strategy profile $\pi$ in which, for each player $i$, $\pi_i$ is an $\epsilon$-BR to $\pi_{-i}$.

A *normal-form game* is a simultaneous-move single-step extensive-form game. An extensive-form game induces a normal-form game in which the legal actions for player $i$ are its deterministic strategies $\times_{s_i \in \mathcal{I}_i} \mathcal{A}_i(s_i)$. These deterministic strategies are called *pure strategies* of the normal-form game. A *mixed strategy* is a distribution over a player's pure strategies.

## 3 ANYTIME DOUBLE ORACLE ALGORITHM (ADO)

Double oracle (DO) (described in B.1) is guaranteed to converge because in the worst case, it will expand all pure strategies, at which point it terminates at a Nash equilibrium (NE). Unfortunately, before convergence, there is no guarantee on the exploitability of the restricted-game NE. In fact, DO can increase exploitability arbitrarily from one iteration to the next.

To see this, consider the game in Figure 1. If both players start with a population consisting only of the first strategy (top row and left column), then the best response for each player is the second strategy, giving that player value 1, for a total exploitability of 2. In the next iteration (Figure 1), when both the first and second strategies are in the population for both players, the restricted-game NE of DO will give probability 1 to the second strategy for each player. This restricted NE has exploitability of 4. In Appendix D, we show empirically that DO does indeed increase exploitability arbitrarily before terminating in this class of games. PSRO inherits this property.

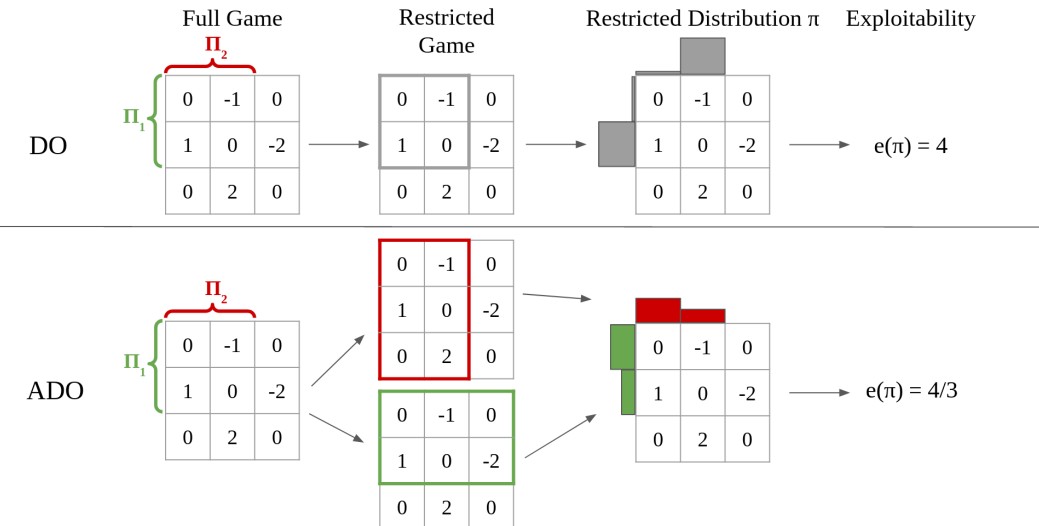

Figure 1: **Top:** In DO, a single restricted game is created and solved in gray. Since this restricted game does not consider strategies outside of the population, it can lead to exploitable restricted distributions. In this example, the DO restricted distribution $\pi$ places all mass on the second strategy, resulting in total exploitability of 4. **Bottom:** Conversely, ADO creates two restricted games where the opponent is unrestricted, player 1's restricted game is shown in green and player 2's restricted game is shown in red. Solving these modified restricted games results in the least-exploitable restricted distributions. In this example, the restricted distribution $\pi$ for ADO puts $\frac{2}{3}$ mass on the first strategy and $\frac{1}{3}$ mass on the second strategy, resulting in the optimal exploitability for this restricted game of $\frac{4}{3}$.

---

**Algorithm 1** Anytime Double Oracle (ADO)

---

**Result:** Nash Equilibrium
**Input:** Initial population $\Pi^0$
**repeat** {for $t = 0, 1, \ldots$}
    **for** $i \in \{1, 2\}$ **do**
        $\pi_i^r \leftarrow$ NE in restricted game $G^i$ (eq. (1))
    **for** $i \in \{1, 2\}$ **do**
        Find a novel best response $\beta_i \leftarrow \mathbb{BR}_i(\pi_{-i}^r)$
        $\Pi_i^{t+1} = \Pi_i^t \cup \{\beta_i\}$
**until** No novel best response exists for either player
**Return:** $\pi^r$

---

In this paper, we introduce *anytime double oracle (ADO)* (Algorithm 1), which is guaranteed not to increase exploitability from one iteration to the next. ADO primarily serves as a foundational element for the development of our subsequent algorithm, APSRO. This foundational role is crucial as it lays the groundwork for APSRO's convergence guarantees. Like DO, ADO maintains a population $\Pi_i^t$ for player $i$ in iteration $t$, and in each iteration computes a Nash equilibrium on a restricted game and adds to each population a best response to the restricted NE. However, unlike DO, ADO creates a different restricted game for each player. The restricted game $G^i$ for player $i$ is created by restricting that player to only play strategies included in their population $\Pi_i$, while the opponent can play *any strategy* in the full game. The game value of $G^i$ for player $i$ is

$$\max_{\pi_i \in \Pi_i} \min_{\pi_{-i}} v_i(\pi_i, \pi_{-i}). \tag{1}$$

The restricted game $G^i$ for player $i$ is then solved for both players to get a NE for each restricted game. We refer to player $i$'s NE strategy in their restricted game as their restricted NE $\pi_i^r$. The restricted NE for player $i$ is the least exploitable mixed strategy supported by player $i$'s population. Note that in large games this restricted game will be prohibitively large to solve and will require approximation with APSRO, introduced later in this paper.

Next, a best response $\beta_i = \mathbb{BR}(\pi^r_{-i})$ is computed for each player $i$ against the restricted-NE mixed strategy of the restricted opponent, and is added to the player's population. If there are multiple best responses, a novel best response $\beta_i \notin \Pi_i$ is chosen that is not currently in that player's population.

ADO is guaranteed to terminate because there are finitely many pure strategies in the original game. When ADO terminates, the restricted NE is a NE in the original game (Proposition 2). Unlike DO, the exploitability of the restricted NE does not increase from iteration to iteration (Proposition 1).

**Proposition 1.** *The exploitability of ADO is monotonically non-increasing.*

*Proof.* All proofs are contained in Appendix H. $\qquad\square$

To illustrate this property of ADO, consider the algorithm dynamics on the DO bad case given in Figure 1. Similar to DO, ADO adds the second strategy to the population in the first iteration. Now, however, instead of taking the second strategy with probability 1 as DO does, ADO solves the restricted game where one player is restricted to the first two strategies and the other is unrestricted and can play any of the three strategies. The Nash equilibrium of this game for the restricted player is to play the first strategy with probability $\frac{2}{3}$ and the second strategy with probability $\frac{1}{3}$. This strategy results in a total exploitability of $\frac{4}{3}$, compared with the DO exploitability of $4$ and the initial ADO exploitability of $2$. In addition to this property of never-increasing exploitability, ADO is guaranteed to converge to a Nash equilibrium, as shown below.

**Proposition 2.** *When ADO terminates, the restricted NE of both players is a Nash equilibrium in the full game.*

## 4 ANYTIME PSRO ALGORITHM (APSRO)

In this section we introduce a scalable extensive-form version of ADO, which we coin *anytime PSRO (APSRO)* (Algorithm 2). Rather than computing the exact NE for each player's ADO restricted game $G^i$, APSRO approximates this solution by simultaneously optimizing each player's restricted distribution $\pi^r_i$ via a regret minimization algorithm against a continuously trained RL best response $\beta_{-i}$. In this work, we update $\pi^r_i$ via the exponential-weight algorithm (Exp3) (Auer et al., 2002) or the Multiplicative Weights Update (MWU) algorithm (Cesa-Bianchi & Lugosi, 2006; Freund & Schapire, 1999).

---

**Algorithm 2** Anytime PSRO

> **Result:** $\epsilon$-Nash Equilibrium
> **Input:** Initial population $\Pi^0$
> **while** Not Terminated $\{t = 0, 1, \ldots\}$ **do**
>     Initialize $\pi^r_i$ to uniform over $\Pi^t_i$ for $i \in \{1, 2\}$
>     Initialize policies $\beta_i$ for $i \in \{1, 2\}$
>     **for** $i \in \{1, 2\}$ **do**
>         **for** $n$ *inner* iterations **do**
>             **for** $m$ iterations **do**
>                 Update policy $\beta_{-i}$ toward $\mathbb{BR}_{-i}(\pi^r_i)$ (e.g. via Q-learning)
>             Update $\pi^r_i$ via regret minimization vs. $\beta_{-i}$ (e.g. via Exp3 or MWU)
>     $\Pi^{t+1}_i = \Pi^t_i \cup \{\beta_i\}$ for $i \in \{1, 2\}$
> **Return:** $\pi^r$

---

Instead of recomputing an exact best response between regret minimization updates, APSRO maintains an approximate best response RL policy $\beta_{-i}$ for each player and updates it for a small number $m$ of steps in each inner-loop iteration. We allow $\beta_{-i}$ to be an approximate best response, and we set the hyperparameter $m$ to a smaller value than may be necessary to fully converge to $\mathbb{BR}_{-i}(\pi^r_i)$. In practice, this trades off the theoretical guarantees of exact best responses with a considerable computational speedup. We include details about the no-regret procedure in Appendix C. The updates to the best response can be made through a variety of algorithms. In this paper we show experiments with updates via tabular Q-learning as well as experiments via the deep reinforcement learning algorithm DDQN (Van Hasselt et al., 2016). Importantly, compared to PSRO, APSRO uses the same

amount of episodes and environment interactions. The only difference is that APSRO changes the restricted distribution dynamically during training via a no-regret procedure.

## 4.1 APSRO THEORY

In this section we present theory for APSRO where we assume the best response is exact in every inner iteration. We show that under this assumption APSRO converges to an approximate Nash equilibrium and never increases exploitability by much. The following proposition shows that APSRO with exact best responses approximately finds the least-exploitable restricted distribution.

**Proposition 3.** *Assume $\beta_{-i} = \mathbb{BR}_{-i}(\pi_i^r)$ in every inner iteration of APSRO. Then APSRO with a regret minimizing algorithm that has regret $R_j$ at inner iteration $j$ will output a policy $\pi^n$ such that $e(\pi^n) \leq \frac{R_n}{n}$.*

By this proposition we know that APSRO with exact best responses will approximately find the least-exploitable restricted distribution for each player in each outer iteration. Since the population grows in every iteration, the least-exploitable distribution of a later iteration is never more exploitable than the least-exploitable distribution of an earlier iteration, simply because exploitability is later minimized over a superset of population mixtures. The following proposition formalizes this intuition.

**Proposition 4.** *Assume APSRO with exact inner-loop best responses runs sufficiently many inner-loop updates in each iteration such that the exploitability in each restricted game is at most $\epsilon$. Then the exploitability of APSRO will never increase by more than $2\epsilon$ from one iteration to the next.*

| 0 | -1 | 0 | 0 | 0 | 1 |
|---|----|---|---|---|---|
| 1 | 0 | -1 | 0 | 0 | 0 |
| 0 | 1 | 0 | -1 | 0 | 0 |
| 0 | 0 | 1 | 0 | -1 | 0 |
| 0 | 0 | 0 | 1 | 0 | -1 |
| -1 | 0 | 0 | 0 | 1 | 0 |

Figure 2: Big RPS Game. Any algorithm that only adds pure best responses, such as common implementations of PSRO or APSRO, will expand all pure strategies before converging.

## 5 SELF-PLAY PSRO

Although ADO and APSRO mitigate increases in exploitability from one iteration to the next by adding to each player's population the pure-strategy best response $\beta_i$ to the opponent's restricted distribution $\pi_{-i}^r$, they are not guaranteed to *decrease* exploitability. $\beta_i$ may not be the myopically optimal pure strategy whose addition to $\Pi_i$ decreases exploitability the most. Moreover, adding mixed strategies can generally reduce exploitability faster than adding pure strategies.

For example, consider the generalized Rock–Paper–Scissors game shown in Figure 2. In this game, the NE mixes equally over all pure strategies. As a result, any DO method that only adds pure strategies, such as common implementations of PSRO, will have to enumerate all pure strategies in the game before supporting the NE.

Ideally, we would like to add a mixed strategy that decreases exploitability the most. A single-iteration objective would then be able to find the strategy such that after it is added to the population and the least-exploitable distribution is computed over this new population, the exploitability of the resulting distribution is the lowest. In this example game, a mixed strategy that mixes over the pure strategies equally is optimal and will lower exploitability more than any pure strategy.

In general, the Nash equilibrium of the original game would be the optimal mixed strategy to add to the population, however finding a Nash equilibrium of the original game is very expensive and is our main goal in the first place.

By trying to add a rough approximation of a Nash equilibrium of the original game to our population, we can still expect to improve our population exploitability a great deal. The closer this new mixed strategy is to being a Nash equilibrium of the original game, the more we would expect it to lower the resulting exploitability of the population.

Motivated by this, we propose Self-Play PSRO, a PSRO method that learns and adds to the population an additional new mixed strategy each iteration. This new strategy is learned by best-responding to the opponent best response via off-policy reinforcement learning in a self play fashion and calculating a mixed-strategy time-average of it. While this self play process won't necessarily produce a Nash

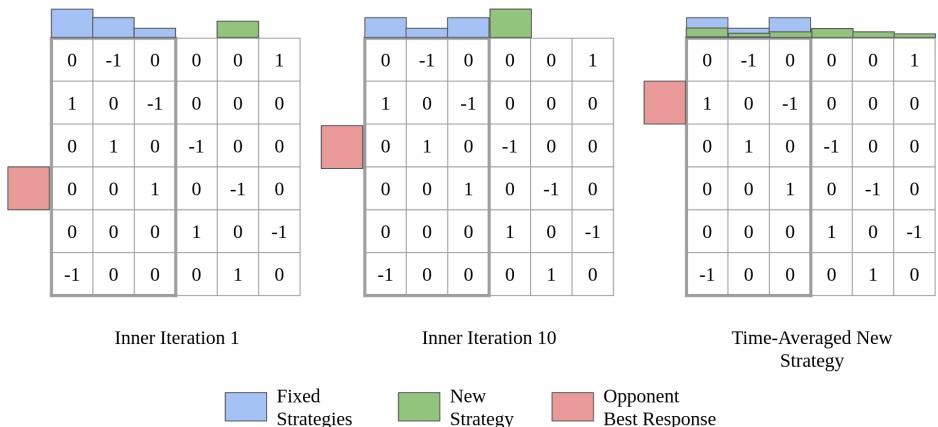

Figure 3: SP-PSRO. In this diagram we show how SP-PSRO works within an iteration from the perspective of the column player. The fixed population is shown in blue and the new strategy is shown in green. Every inner iteration, three things happen. (1) The opponent best response updates toward a best response to the current distribution over both the fixed population and the new strategy. (2) The new strategy updates toward a best response against the opponent best response. (3) The restricted distribution updates via no regret against the opponent best response. In the final iteration, the time-average of the new strategy and the player's best response to the opponent's restricted distribution (which is trained in a symmetric manner) are added to the population, and the cycle starts again.

equilibrium, this additional strategy can serve as an inexpensive heuristic approximation of one to dramatically reduce our population's exploitability, especially in earlier PSRO iterations.

---

**Algorithm 3** Self-Play PSRO

---

**Result:** Approximate Nash Equilibrium
**Input:** Initial population $\Pi^0$
**while** Not Terminated $\{t = 0, 1, \ldots\}$ **do**
   **for** $i \in \{1, 2\}$ **do**
      Initialize new strategy $\nu_i$ arbitrarily
      Initialize $\pi_i^r$ to uniform over $\Pi_i^t \cup \{\nu_i\}$
      **for** $n$ iterations **do**
         **for** $m$ iterations **do**
            Update policy $\beta_{-i}$ toward $\mathbb{BR}_{-i}(\pi_i^r)$ (e.g. via Q-Learning)
            Update new strategy $\nu_i$ toward $\mathbb{BR}_i(\beta_{-i})$ (e.g. via Q-Learning)
         Update $\pi_i^r$ via regret minimization vs. $\beta_{-i}$ (e.g. via Exp3 or MWU)
      $\Pi_i^{t+1} = \Pi_i^t \cup \{\beta_i, \bar{\nu}_i\}$ for $i \in \{1, 2\}$
**Return:** $\pi^r$

---

SP-PSRO works by maintaining a restricted distribution $\pi_i^r$ over a population. Unlike PSRO, where $\pi_i^r$ is the NE of the restricted game, SP-PSRO trains $\pi_i^r$ in the same way as in APSRO, via regret minimization. In addition, at the beginning of each iteration, a new strategy $\nu_i$ is initialized and added to the population.

During an iteration, three training processes unfold concurrently. First, as in APSRO, the opponent's best response $\beta_i$ takes multiple update steps toward a best response to the current restricted distribution $\mathbb{BR}_{-i}(\pi_i^r)$. Second, the new strategy $\nu_i$ is updated toward a best response to the opponent best response $\mathbb{BR}_i(\beta_{-i})$. Third, the restricted distribution $\pi_i^r$ is trained via regret minimization; this includes updating the probability of the new population strategy $\nu_i$, even as $\nu_i$ is also trained. This procedure can be thought of a form of self-play, in which the new strategy is updating against the opponent best response, while the opponent best response is updating against the restricted distribution, which also contains the new strategy. When the iteration is finished, the time-average $\bar{\nu}_i$ of $\nu_i$ is added to the population. We include further details on SP-PSRO in Appendix L.3.

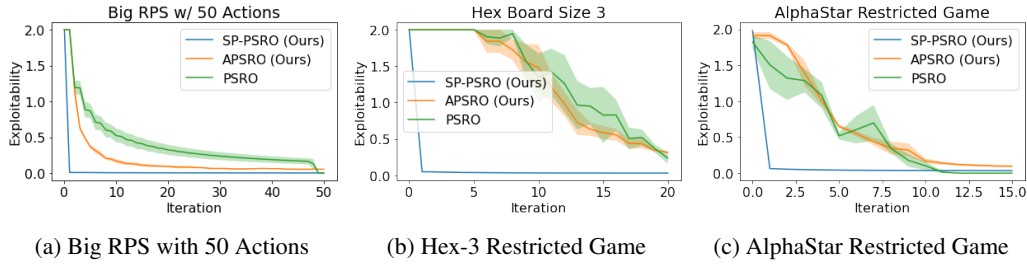

Figure 4: Normal-form games

Averaging over the updates of $\nu_i$ can be accomplished by checkpointing the policy over time and uniformly sampling checkpoints, or by training a neural network to distill a buffer of experience generated by $\nu_i$ as it trains. Since the new strategy is trained via off-policy reinforcement learning, SP-PSRO uses the same amount of environment experience as APSRO, but does require more compute to train the new network. Additionally, since it still adds best responses $\beta$, similar to APSRO and PSRO, it will also converge to an optimal population that supports a NE.

## 6 EXPERIMENTS

### 6.1 NORMAL FORM EXPERIMENTS

In this section we describe experiments on normal form games. To emulate the process of a strategy $\pi$ learning a best response to another policy $\pi'$, in every inner loop iteration $t$ we update $\pi$ by the following learning rule: $\pi_{t+1} = (1 - \lambda)\pi_t + \lambda \times \mathbb{BR}_i(\pi')$. We show three normal form games. The first, described in Figure 4a, is a large generalized Rock–Paper–Scissors game. The second is a a Hex restricted game (Perez-Nieves et al., 2021). The third game is the final restricted game of the AlphaStar population (Vinyals et al., 2019). More normal form games are included in Appendix E. As shown in Figure 4, SP-PSRO vastly outperforms both PSRO and APSRO. Note that APSRO and SP-PSRO only reach an $\epsilon$-NE because they use a finite number of regret minimization updates to determine the restricted distribution, while PSRO is able to exactly compute a NE. We have included further details in the Appendix.

### 6.2 TABULAR EXPERIMENTS

We evaluated SP-PSRO with tabular methods in a variety of games. We applied tabular SP-PSRO to the domains of Leduc Poker (9,457 states), a tiny version of Battleship (1,573 states), and 4x-Repeated Rock Paper Scissors (9,841 states). The experiments used game implementations and tools from the OpenSpiel library (Lanctot et al., 2019).

In extensive form tabular experiments, the new population strategy $\nu_i$ and the best response $\beta_{-i}$ are represented by tabular Q-learning agents. When training the Q-learning agent for $\beta_{-i}$, experience from the same episodes is also used to train the agent for $\nu_i$ in an off-policy manner. The tabular Q-learning agents are $\epsilon$-greedy, and we use a constant value of $\epsilon$ for both agents. Because experience for $\beta_{-i}$ and $\nu_i$ is shared from the same episodes, experience is collected against $\epsilon$-greedy versions of some opponent policies. Compared to collecting separate episodes for each player, we found that using the same episodes to train policies for both players despite small amounts of action exploration reduces the required sample complexity by two without affecting performance very much. In these 3 games, we collect the same amount of experience per iteration for PSRO, APSRO, and SP-PSRO.

Similar to our normal form results, we find that APSRO does not increase exploitability by much from one iteration to the next and SP-PSRO drastically reduces exploitability compared to baselines. Interestingly, in tiny battleship, the exploitability of APSRO had higher variance compared to that of PSRO. We hypothesize that this is due to the APSRO iterations not being long enough for the no-regret process to converge. We have included further details in the Appendix.

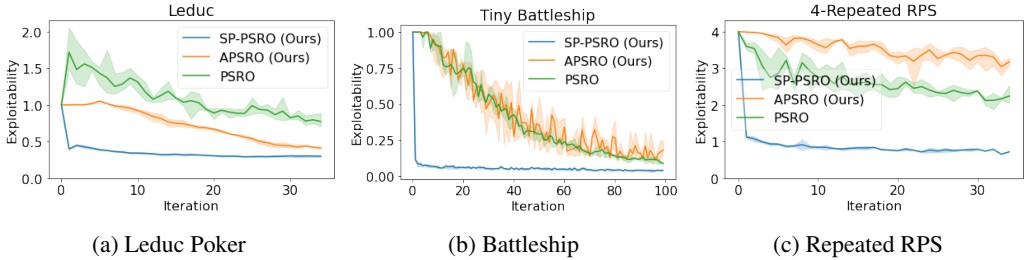

Figure 5: Extensive-form games with tabular Q-learning best responses

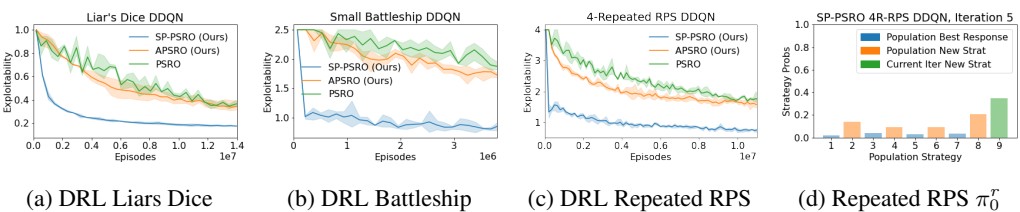

Figure 6: Extensive-form games with DDQN best responses

SP-PSRO outperforms APSRO and PSRO in each of the three games: Leduc Poker (Figure 5a), the small Battleship game (Figure 5b), and 4-repeated Rock Paper Scissors (Figure 5c). In each game, we see a drastic improvement in performance starting in the first iteration.

## 6.3 DEEP REINFORCEMENT LEARNING EXPERIMENTS

When using deep reinforcement learning best-response operators with DDQN (Van Hasselt et al., 2016), SP-PSRO outperforms APSRO and PSRO in terms of sample efficiency (Figure 6). Tested on Liar's Dice, a small version of Battleship, and 4x-Repeated RPS, SP-PSRO sees a significant improvement against other baselines in early-iteration exploitability. This early exploitability advantage seen by SP-PSRO is especially present in repeated RPS (Figure 6c), where the relative performance seen with deep RL methods roughly matches that of tabular methods. Examining a player's final restricted distribution after 5 iterations of SP-PSRO in repeated RPS (Figure 6d), we also see that the time-averaged new strategies have significantly more support than the standard best-responses, demonstrating their contribution towards exploitability improvements.

## 7 FUTURE WORK

SP-PSRO opens up exciting connections to the literature regarding learning approximate Nash equilibria in large games. In particular, although we introduce an unprincipled self play method for approximating a Nash equilibrium, future work can find better ways of creating a new strategy that will better approximate a Nash equilibrium and therefore result in lower exploitability every iteration. For example, the data collected via the the opponent best response training against the restricted distribution can be used in a Monte-Carlo CFR-type algorithm to minimize regret on information sets visited during training. These directions also open up the possibility of deriving the first regret bounds for double oracle algorithms that do not rely on the size of the effective pure strategy set (Dinh et al., 2021). It also introduces the possibility of combining the deep reinforcement learning from the best response with methods based on deep CFR. For example, perhaps the Q networks learned from the best responses can be used to minimize regret for the new strategy. Finally, our algorithm is a normal-form algorithm in that it mixes at the root of the game tree. McAleer et al. (2021) showed that this can be exponentially bad in the worst case, and introduced tabular (XDO) and deep (NXDO) algorithms to fix this problem. An interesting future direction is combining SP-PSRO with XDO and NXDO.

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

# A    LIMITATIONS

One limitation of SP-PSRO is that if the new strategies happen to not be useful, including the new strategy can hurt the performance of the restricted distribution. This is primarily because it is harder to learn a no-regret distribution when one of the arms is changing, and secondly because including more actions (strategies) makes it harder for the no-regret algorithm as well. A related limitation of SP-PSRO is that because including the new strategy makes it harder to learn the restricted distribution, we find that SP-PSRO tends to plateau higher than APSRO and can even slightly increase exploitability. To mitigate this, one can switch over to APSRO after some iterations, but we have not introduced a principled method of determining when is a good time to switch. A third limitation of our method is that extra compute needs to be used to train the new strategy $\nu$. Also, if the average strategy is computed via supervised learning on a replay buffer of experience, this adds additional memory requirements to the algorithm. Also, while ideas from this paper can potentially be applied to solution concepts beyond Nash equilibrium, we did not explore those directions in this paper.

# B    RELATED WORK

Many recent works study the intersection of reinforcement learning and game theory. QPG (Srinivasan et al., 2018) is an algorithm based on policy gradient that empirically converges to a NE when the learning rate is annealed. NeuRD (Hennes et al., 2020), Magnetic Mirror Descent (Sokota et al., 2022), and F-FoReL (Perolat et al., 2021) approximate replicator dynamics, mirror descent, and follow the regularized leader, respectively, with policy gradients. DeepNash, which is based on F-FoReL and NeuRD has achieved expert level performance at Stratego (Perolat et al., 2022). Markov games generalize MDPs where players take simultaneous actions and observe the ground state of the game. Recent literature has shown that reinforcement learning algorithms converge to Nash equilibrium in two-player zero-sum Markov games (Brafman & Tennenholtz, 2002; Wei et al., 2017; Perolat et al., 2018; Xie et al., 2020; Daskalakis et al., 2020; Jin et al., 2021) and in multi-player general-sum Markov potential games (Leonardos et al., 2021; Mguni et al., 2021; Fox et al., 2022; Zhang et al., 2021; Ding et al., 2022). Deep methods based on CFR (McAleer et al., 2022; Steinberger et al., 2020; Brown et al., 2019) are another promising direction for scaling to large games. In this work we focus on a different set of deep RL algorithms for games based on PSRO. Advances made to PSRO can potentially be combined with the above methods via XDO (McAleer et al., 2021).

## B.1    DOUBLE ORACLE (DO) AND POLICY SPACE RESPONSE ORACLES (PSRO)

Double oracle (McMahan et al., 2003) is an algorithm for finding a Nash equilibrium (NE) in normal-form games. The algorithm works by keeping a population of strategies $\Pi^t$ at time $t$. In each iteration, a NE $\pi^{*,t}$ is computed for the game restricted to strategies in $\Pi^t$. Then, a best response to this restricted NE for each player $\mathbb{BR}_i(\pi^{*,t}_{-i})$ is computed and added to the population $\Pi^{t+1}_i = \Pi^t_i \cup \{\mathbb{BR}_i(\pi^{*,t}_{-i})\}$ for $i \in \{1,2\}$. Although in the worst case DO must add all pure strategies, in many games DO empirically terminates early and outperforms alternative approaches.

---

**Algorithm 4** Policy Space Response Oracle (PSRO) (Lanctot et al., 2017)

---

   **Result:** Nash Equilibrium
   **Input:** Initial population $\Pi^0$
   **repeat** {for $t = 0, 1, \ldots$}
      $\pi^r \leftarrow$ NE in game restricted to strategies in $\Pi^t$
      **for** $i \in \{1,2\}$ **do**
         **for** $m$ iterations **do**
            Update policy $\beta_{-i}$ toward $\mathbb{BR}_{-i}(\pi^r_i)$
      $\Pi^{t+1}_i \leftarrow \Pi^t_i \cup \{\beta_i\}$ for $i \in \{1,2\}$
   **until** No novel best response exists for either player
   **Return:** $\pi^r$

---

Policy-Space Response Oracles (PSRO) (Lanctot et al., 2017) scales DO to large games by using reinforcement learning to approximate a best response. The restricted-game NE is computed on the restricted game matrix $U^\Pi$, generated by having each policy in the population $\Pi$ play each opponent

policy and tracking average utility in a $\Pi_1 \times \Pi_2$ payoff matrix (Wellman, 2006). PSRO is described in Algorithm 4.

Several methods related to PSRO have been published in recent years. AlphaStar (Vinyals et al., 2019) trains a population of policies through a procedure that is somewhat similar to PSRO. AlphaStar also uses some elements of self-play when constructing its population, and outputs a population-restricted NE at test time. NXDO (McAleer et al., 2021) iteratively adds reinforcement learning policies to a population but solves an extensive-form restricted game, which has been shown to be more efficient than solving a matrix-form restricted game as in PSRO. P2SRO (McAleer et al., 2020) parallelizes PSRO with convergence guarantees. Other work has looked at incorporating diversity (Liu et al., 2021; Perez-Nieves et al., 2021) in the best response objective. However, since the best response is still pure in most implementations, these methods suffer from the same problems of PSRO and APSRO as previously described. Other methods generalize PSRO to more players (Muller et al., 2020; Marris et al., 2021), and meta-learn the restricted-NE population distribution (Feng et al., 2021). Slumbers et al. (2022) propose a PSRO-like approach for learning risk-averse equilibria.

## B.2 MINIMUM-REGRET CONSTRAINED PROFILE

The concept of finding a low-exploitability distribution in a restricted game was also explored in Jordan et al. (2010) and Wang et al. (2022), which define the *minimum regret constrained profile* as the distribution over a restricted population that achieves the lowest exploitability. In this paper, we use the term *least-exploitable restricted distribution* for the same concept because we believe it better emphasizes the fact that this restricted distribution is the least-exploitable distribution over the restricted population.

## B.3 RANGE OF SKILL ALGORITHM (ROS)

This paper presents ADO, which can be viewed as a modification to the *range of skill (ROS)* algorithm introduced by Zinkevich et al. (2007) and further explored in Hansen et al. (2008). ROS is a variant of the DO algorithm that likewise produces a series of restricted games by iteratively adding new strategies. As in ADO, ROS defines in each iteration two separate restricted games, in each of which one player is restricted to play strategies in their population while the other player is unrestricted. The similarity between ROS and ADO continues in computing a Nash equilibrium strategy profile for each restricted game, such that the restricted player's strategy is the least-exploitable restricted distribution.

However, ADO and ROS differ in the strategy that they then add to the unrestricted player's population. ADO adds a best response to the restricted distribution, while ROS adds a strategy that is part of the unrestricted player's NE strategy in the restricted game.

This difference proves crucial when scaling up to large games. In large games, solving the restricted game where one player is unrestricted is infeasible and as a result methods based on ROS cannot scale to large games. Alternatively, since ADO only adds best responses, it naturally scales to large games via APSRO where the best responses are learned through RL.

Additionally, while ROS, like ADO, decreases exploitability monotonically and performs well in practice, the only known convergence guarantees for ROS are asymptotic with a convergence rate that is exponential in the size of the game (Hansen et al., 2008). In contrast, ADO is guaranteed to converge in a number of iterations at most the number of pure strategies in a game.

## C   APSRO AND SP-PSRO NO-REGRET ALGORITHMS

In the inner loops of APSRO and SP-PSRO, we update $\pi_i^r$ via regret minimization against $\beta_{-i}$. To do this, we use two different no-regret algorithms, the exponential-weight algorithm (Exp3) (Auer et al., 2002) for tabular extensive-form experiments and the Multiplicative Weights Update (MWU) algorithm (Cesa-Bianchi & Lugosi, 2006; Freund & Schapire, 1999) for deep RL extensive-form experiments.

### C.1   EXP3

Exp3, shown in Algorithm 5 is an adversarial bandit method that has sublinear regret. We use Exp3 as our no-regret restricted game solver in tabular APSRO and SP-PSRO experiments. We perform batches of multiple RL updates in alternation with batches of multiple Exp3 updates. Exploitability is evaluated using the final Exp3 sampling distribution $P_n$ for each APSRO/SP-PSRO iteration.

---

**Algorithm 5** Exp3

---

**Input:** $n$ iterations, $k$ actions, parameter $\gamma$
Initialize cumulative rewards $\hat{S}_0 = (0, 0, ...0)$
**for** $t = 1, ..., n$ **do**
    Calculate the sampling distribution $P_{t,i}$: $P_{t,i} = (1 - \gamma)\frac{exp(\gamma\hat{S}_{t-1,i}/k)}{\sum_{j=1}^{k} exp(\gamma\hat{S}_{t-1,j}/k)} + \frac{\gamma}{k}$ for each $i \in [1..k]$
    Sample action $A_t \sim P_t$ and observe reward $X_t$
    Calculate $\hat{S}_{t,i}$: $\hat{S}_{t,i} = \hat{S}_{t-1,i} + \frac{X_t\mathbb{1}\{A_t=i\}}{P_{t,i}}$

---

### C.2   MULTIPLICATIVE WEIGHTS UPDATE

The Multiplicative Weights Update (MWU) algorithm is an online learning method which converges in time-average to Nash equilibrium (Cesa-Bianchi & Lugosi, 2006; Freund & Schapire, 1999). We use MWU, shown in Algorithm 6 as our no-regret restricted game solver in deep RL APSRO and SP-PSRO experiments. For all games, we perform a metasolver update once after multiple iterations of our RL best response algorithm. When measuring exploitability, we evaluate the final output $\pi^r$ as the time-average of the MWU sampling distribution $P_t$ over a single APSRO/SP-PSRO iteration.

---

**Algorithm 6** Multiplicative Weights Update

---

**Input:** $n$ iterations, $k$ actions, learning rate $\eta$
Initialize cumulative rewards $\hat{S}_0 = (0, 0, ...0)$
**for** $t = 1, ..., n$ **do**
    Calculate the sampling distribution $P_{t,i}$: $P_{t,i} = \frac{exp(\eta\hat{S}_{t-1,i})}{\sum_{j=1}^{k} exp(\eta\hat{S}_{t-1,j})}$ for each $i \in [1..k]$
    Observe reward $X_{t,i}$ for each action $i \in [1..k]$
    **for** $i = 1, ...k$ **do**
        Calculate $\hat{S}_{t,i}$ : $\hat{S}_{t,i} = \hat{S}_{t-1,i} + X_{t,i}$

---

# D DOUBLE ORACLE VS ANYTIME DOUBLE ORACLE

To demonstrate how DO can increase exploitability in every iteration except the last, consider a generalization of the game presented in Figure 1 where all values are 0, except if the row index $r$ is one more than the column index $c$, in which case the value for the row player is $\sum_{i=0}^{r} 2^i + 2i$, or if the column index $c$ is one more than the row index $r$, in which case the value for the row player is $\sum_{i=0}^{c} -2^i + 2i$. We plot the performance of DO and ADO in this game with 10 actions in Figure 7a and show that DO increases exploitability in every iteration except the last, while ADO does not increase exploitability.

Figure 7b also compares DO and ADO on random normal-form games with 500 actions. We see that ADO greatly outperforms DO and tends not to increase exploitability. To create random normal-form games, we sample payoff values from Uniform(0,1).

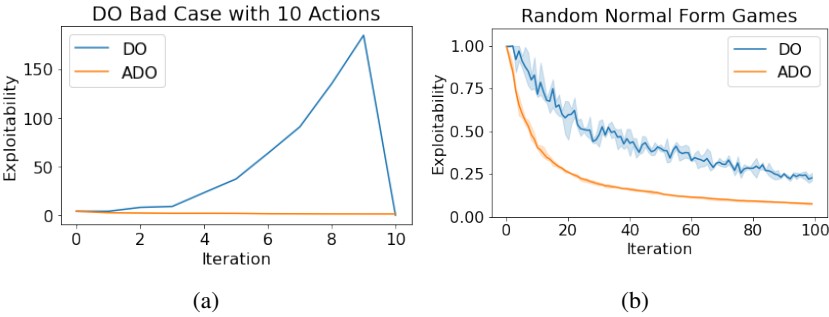

(a)                                     (b)

Figure 7: (a) DO can arbitrarily increase exploitability before convergence, whereas ADO monotonically decreases exploitability and guarantees convergence to a Nash equilibrium. (b) Random Normal Form Games with 500 Actions

# E    ADDITIONAL NORMAL-FORM EXPERIMENTS

In this section we report additional normal form game experiments. All games in this section are from Perez-Nieves et al. (2021). Note that the zeroth iteration is not included in the plots. Similar to the main results in the paper we find that SP-PSRO achieves much lower exploitability than existing PSRO based methods and does so much faster, across all games studied. We include an ablation, labeled SP-PSRO Not Anytime, that is the same as SP-PSRO in that it trains a new strategy to be a best response to the opponent best response, but unlike SP-PSRO does not update the restricted distribution via no-regret as in anytime PSRO. As shown in the figures, anytime PSRO is a crucial piece of SP-PSRO, and excluding this aspect results in much worse performance. We find that when anytime PSRO is excluded, the opponent best response will be best responding to a static opponent, and the best response to this best response will tend to be a pure strategy. As a result, we do not get to explore the strategy space, and the average new strategy will simply be another pure strategy. In some games we see that SP-PSRO Not Anytime and PSRO converge to lower exploitability than SP-PSRO and anytime PSRO. This is because SP-PSRO Not Anytime and PSRO both use exact meta-solvers, which return the exact Nash equilibrium upon convergence, while SP-PSRO and APSRO use an approximate no-regret procedure to find the least-exploitable restricted distribution.

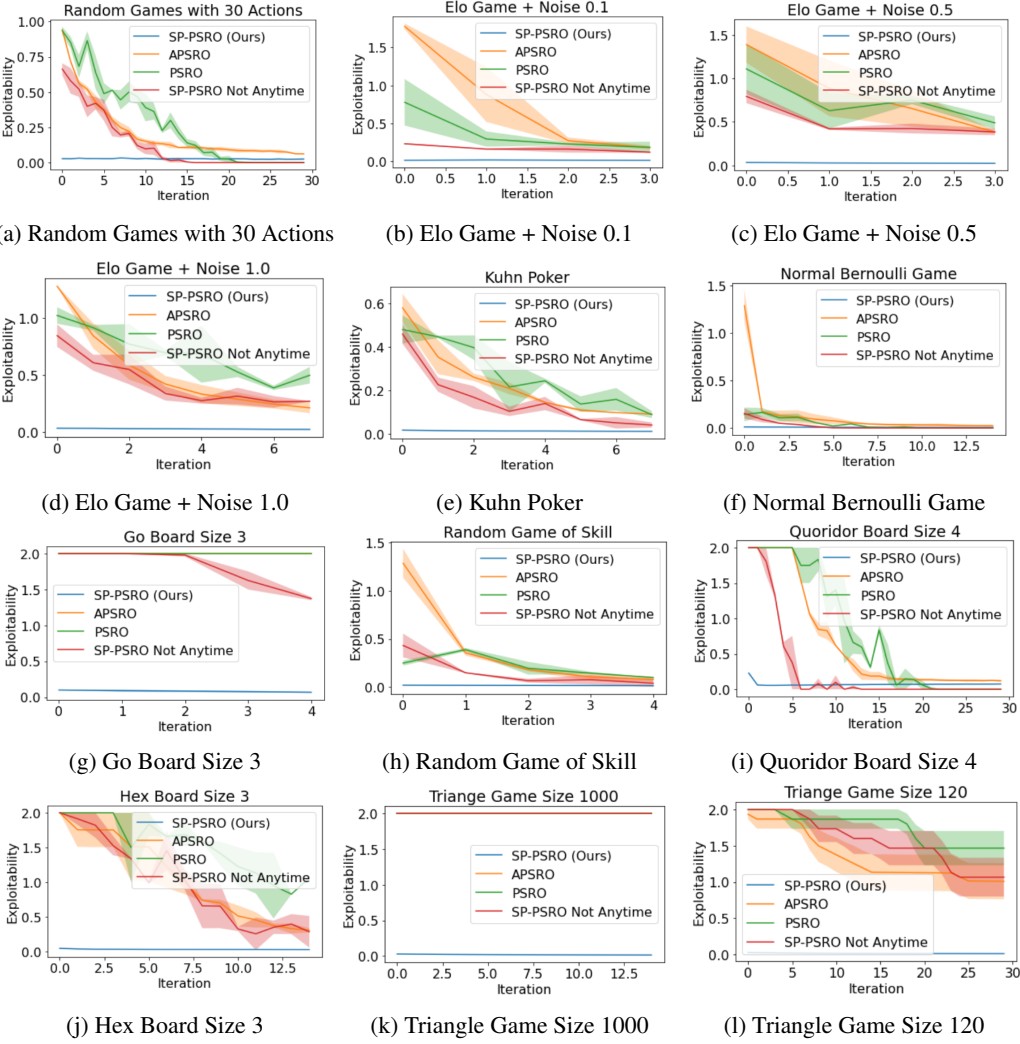

Figure 8: Additional Normal-Form Game Experiments

### E.1 NORMAL-FORM ABLATIONS

In this section we run ablations with different levels of $\lambda$ for updating the best response. As shown in these experiments, the relative performance does not change much.

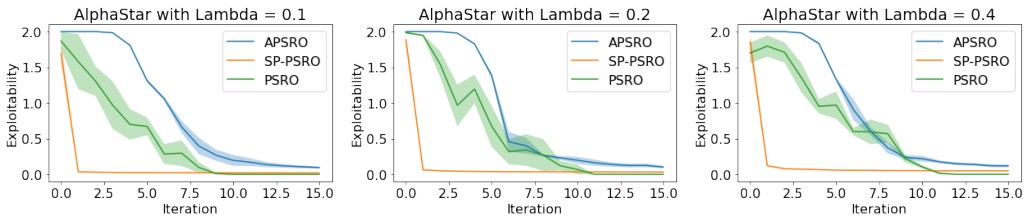

Figure 9: Changing the $\lambda$ parameter does not change the results very much.

### E.2 PSRO-RECTIFIED NASH

We additionally compare to PSRO-Rectified Nash as a baseline. Since we only focus on methods that are guaranteed to converge to Nash equilibrium, and since PSRO Rectified Nash has been shown not to converge to NE McAleer et al. (2020), we do not include this baseline in the main text. As shown here, it is able to outperform DO while underperforming ADO in random games.

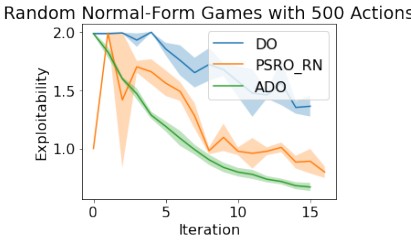

Figure 10: ADO outperforms PSRO-Rectified Nash on random normal form games.

## F ADDITIONAL TABULAR EXPERIMENT ON GOOFSPIEL

We additionally compare SP-PSRO and APSRO against PSRO on Goofspiel with tabular Q-learning best responses. As shown in figure 11, like in other games, SP-PSRO heavily outperforms APSRO and PSRO in early-iteration exploitability. APSRO outperforms PSRO in early-iteration exploitability but not by as much as SP-PSRO. The high iteration-over-iteration variance and final exploitability that APSRO exhibits is likely due to APSRO's approximate solution to its restricted game. The accuracy of APSRO's no-regret solution for $\pi^r$ is dependent on the number of inner-loop iterations and the ratio of approximate best-response updates to no-regret updates. The optimal values for each game may vary, and these hyperparameters were kept constant across all games on tabular experiments.

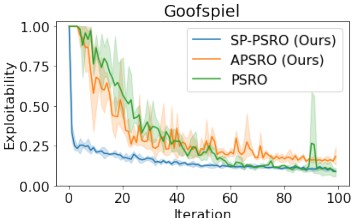

Figure 11: Extensive-form Goofspiel with tabular Q-learning best responses

# G  ADDITIONAL DEEP RL EXPERIMENTS

## G.1  LAST-ITERATE SP-PSRO

We compare an alternate last-iterate version of SP-PSRO against the default SP-PSRO method and other baselines in Figure 12. In the SP-PSRO last-iterate variant, we add the pure-strategy weights of $\nu_i$ in its final RL iteration to the population rather than calculating and adding the time average $\bar{\nu}_i$. Exploitability is also calculated using $\nu_i$ rather than $\bar{\nu}_i$. SP-PSRO last-iterate improves upon APSRO and PSRO due to the additional, potentially useful, population policy. However, $\nu_i$ is less able to roughly approximate a NE because it represents a single pure-strategy approximate best-response to $\beta_{-i}$ rather than a mixture of multiple approximate best-responses distributed across each time-slice of an SP-PSRO iteration. Because of this, we still see an additional gain in exploitability versus sample-efficiency when transitioning from SP-PSRO last-iterate to the default time-average version of SP-PSRO.

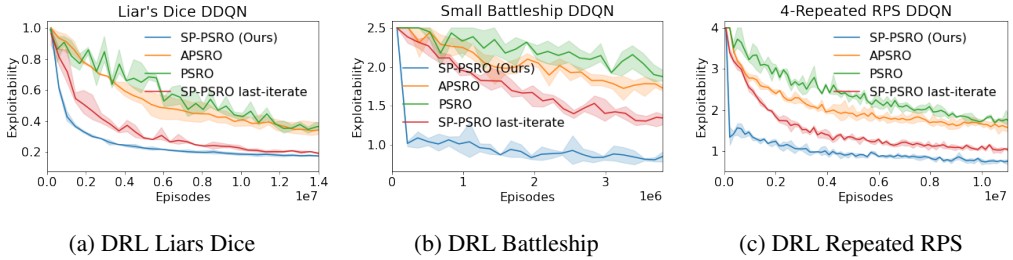

(a) DRL Liars Dice          (b) DRL Battleship          (c) DRL Repeated RPS

Figure 12: SP-PSRO last-iterate on extensive-form games with DDQN best responses

## G.2  SP-PSRO NOT ANYTIME

Similar to the additional normal-form experiment in Appendix E, we compare against a variant of SP-PSRO denoted as SP-PSRO Not Anytime. In this variant, the Nash equilibrium for the restricted game over $\Pi^t$ is used as $\pi^r$, similar to PSRO. By ablating the no-regret restricted-game solver, SP-PSRO Not Anytime has phases in which it dramatically increases in exploitability as iterations progress in Small Battleship and 4-Repeated RPS. We note that because the training dynamics of $\beta_i$ and $\nu_i$ are affected by the choice of $\pi^r$, it is possible that SP-PSRO Not Anytime will learn less effective population strategies. We find that SP-PSRO Not Anytime outperforms PSRO and APSRO but underperforms SP-PSRO. This lends evidence to the hypothesis that the success of SP-PSRO is due to both the new strategy and the anytime regret-minimization procedure working together.

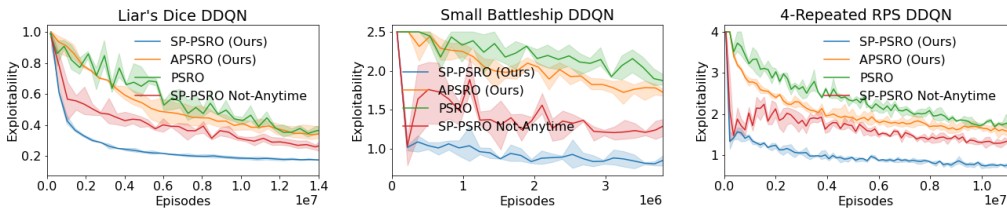

Figure 13: Additional ablation demonstrating the exploitability of a Not Anytime variant of SP-PSRO in deep RL experiments.

## G.3  HEAD-TO-HEAD VS RANDOM PERFORMANCE

For each primary algorithm tested, we compare the performance of $\pi^r$ against an opponent that selects actions randomly. In games where playing randomly isn't an NE like Liar's Dice and Small Battleship, performance against a fixed random opponent is roughly anticorrelated with exploitability. Each method performs equally against random in 4-Repeated RPS because playing randomly aligns with the NE in this game, so a random opponent is not exploitable.

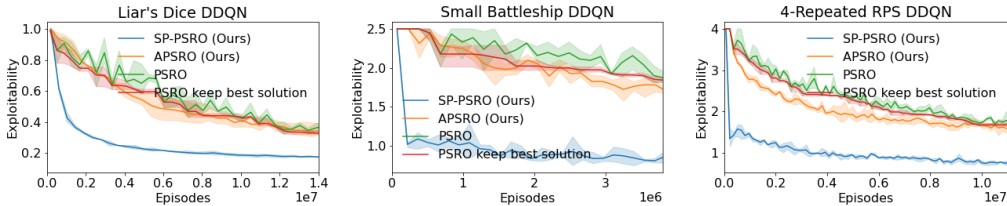

Figure 14: Performance vs Random for each Deep RL method as a function of experience collected on Liar's Dice, Small Battleship, and 4-Repeated RPS.

## G.4 PSRO KEEP BEST NE

As an additional baseline, we evaluate a variant of PSRO where one keeps the least-exploitable meta-NE seen so far. By construction, this algorithm does not increase exploitability, but still performs slightly worse than APSRO.

Figure 15: Exploitability of methods including a baseline where PSRO keeps the best meta-NE each iteration.

# H    PROOFS

## H.1    PROOF OF PROPOSITION 1

*Proof.* Let $\pi^t$ be the restricted NE in ADO at iteration $t$. Then for player $i$, since $\Pi_i^t \subseteq \Pi_i^{t+1}$

$$
\begin{aligned}
v_i(\pi_i^t, \mathbb{BR}_{-i}(\pi_i^t)) &= \max_{\pi_i \in \Pi_i^t} \min_{\pi_{-i}} v_i(\pi_i, \pi_{-i}) \\
&\leq \max_{\pi_i \in \Pi_i^{t+1}} \min_{\pi_{-i}} v_i(\pi_i, \pi_{-i}) \\
&= v_i(\pi_i^{t+1}, \mathbb{BR}_{-i}(\pi_i^{t+1})).
\end{aligned}
\tag{2}
$$

Since each player's value is monotonically non-decreasing, the value of the best response is non-increasing, and the exploitability of $\pi^t$ is also non-increasing:

$$
\begin{aligned}
e(\pi^{t+1}) &:= -\sum_i v_i(\pi_i^{t+1}, \mathbb{BR}_{-i}(\pi_i^{t+1})) \\
&\leq -\sum_i v_i(\pi_i^t, \mathbb{BR}_{-i}(\pi_i^t)) = e(\pi^t).
\end{aligned}
\tag{3}
$$

$\square$

## H.2    PROOF OF PROPOSITION 2

*Proof.* Let $(\pi_1^r, \pi_2')$ and $(\pi_1', \pi_2^r)$ be the NE in the restricted games $G^1$ and $G^2$ for player 1 and 2, respectively, at termination. If $\pi_1'$ or $\pi_2'$ have support outside the population, ADO would not terminate, because there would exist another novel best response. Hence, at termination, the support of both $\pi_1'$ and $\pi_2'$ are inside the population and they are feasible for their respective player's restricted game. Then

$$
\begin{aligned}
v_1(\pi_1^r, \pi_2^r) &\leq v_1(\pi_1', \pi_2^r) \\
&\leq v_1(\pi_1', \pi_2') \\
&\leq v_1(\pi_1^r, \pi_2') \\
&\leq v_1(\pi_1^r, \pi_2^r).
\end{aligned}
\tag{4}
$$

The four inequalities follow, in order, because: (a) player 1 doesn't want to deviate from $\pi_1'$ to $\pi_1^r$ in $G^2$; (b) player 2 doesn't want to deviate from $\pi_2^r$ to $\pi_2'$ in $G^2$; (c) player 1 doesn't want to deviate from $\pi_1^r$ to $\pi_1'$ in $G^1$; and (d) player 2 doesn't want to deviate from $\pi_2'$ to $\pi_2^r$ in $G^1$.

Therefore, $v_1(\pi_1^r, \pi_2^r) = v_1(\pi_1', \pi_2^r)$ which implies that player 1 has no incentive to deviate from $\pi_1^r$ to $\pi_1'$ or any other strategy against $\pi_2^r$. A symmetric argument holds for player 2, implying that $\pi^r$ is a Nash equilibrium in the full game. $\square$

## H.3    PROOF OF PROPOSITION 3

*Proof.* The proof follows the same argument as used in Theorem 3 of Johanson et al. (2012). By a folk theorem of game theory, we know that if two algorithms with regret $R_j^0$ and $R_j^1$ play each other in self play in a two-player zero-sum normal-form game, their average joint strategy at time $n$, $\pi^n$, will have exploitability less than or equal to $\frac{R_n^0 + R_n^1}{n}$. So it remains to show that the best responder has negative regret. To see this, note that changing the action at all time steps to be any one action for the best responder would be no better than the actual action, which was the best response. $\square$

## H.4    PROOF OF PROPOSITION 4

*Proof.* Let $\pi^t$ be the restricted NE of APSRO at iteration $t$. Then

$$
v_i(\pi_i^t, \mathbb{BR}_{-i}(\pi_i^t)) \leq \max_{\pi_i \in \Pi_i^t} \min_{\pi_{-i}} v_i(\pi_i, \pi_{-i}) \tag{5}
$$

$$
\leq \max_{\pi_i \in \Pi_i^{t+1}} \min_{\pi_{-i}} v_i(\pi_i, \pi_{-i}) \tag{6}
$$

$$
\leq v_i(\pi_i^{t+1}, \mathbb{BR}_{-i}(\pi_i^{t+1})) + \epsilon, \tag{7}
$$

where (5) follows from $\pi_i^t$ being feasible for player $i$'s restricted game in iteration $t$, whose value is on the right-hand side; (6) from population monotonicity $\Pi_i^t \subseteq \Pi_i^{t+1}$; and (7) from the $\epsilon$-exploitability of $\pi_i^{t+1}$ in the restricted game in iteration $t+1$. The proposition now follows from

$$e(\pi^{t+1}) := -\sum_i v_i(\pi_i^{t+1}, \mathbb{BR}_{-i}(\pi_i^{t+1})) \tag{8}$$

$$\leq -\sum_i \left( v_i(\pi_i^t, \mathbb{BR}_{-i}(\pi_i^t)) - \epsilon \right) = e(\pi^t) + 2\epsilon.$$

$\square$

# I ANALYSIS OF RESTRICTED GAME DISTRIBUTIONS

In this section, we qualitatively analyze the learned restricted game distributions $\pi^r$ over the populations of PSRO, APSRO, and SP-PSRO. We examine checkpoints from a single seed of each of our deep reinforcement learning experiments in section 6.3 on Liar's Dice, Small Battleship, and 4-Repeated RPS. Given these checkpoints, we observe the first player's learned restricted distribution $\pi_0^r$ for the first 7 iterations of each algorithm. For PSRO, $\pi^r$ is the NE to the restricted game, whereas in APSRO and SP-PSRO, $\pi^r$ is an approximate no-regret solution for each player to the restricted game with an unrestricted opponent. In each iteration, PSRO and APSRO add a single new best response $\beta_{-i}$ for each player $i$ to $\Pi_{-i}$. In addition to $\beta_{-i}$, SP-PSRO adds a time-average of the new-strategy $\bar{\nu}_i$ to $\Pi_i$ in each iteration. These distributions $\pi_0^r$ over the first player's population are shown for Liar's Dice in figure 16, for Small Battleship in figure 17, and for 4-Repeated RPS in figure 18.

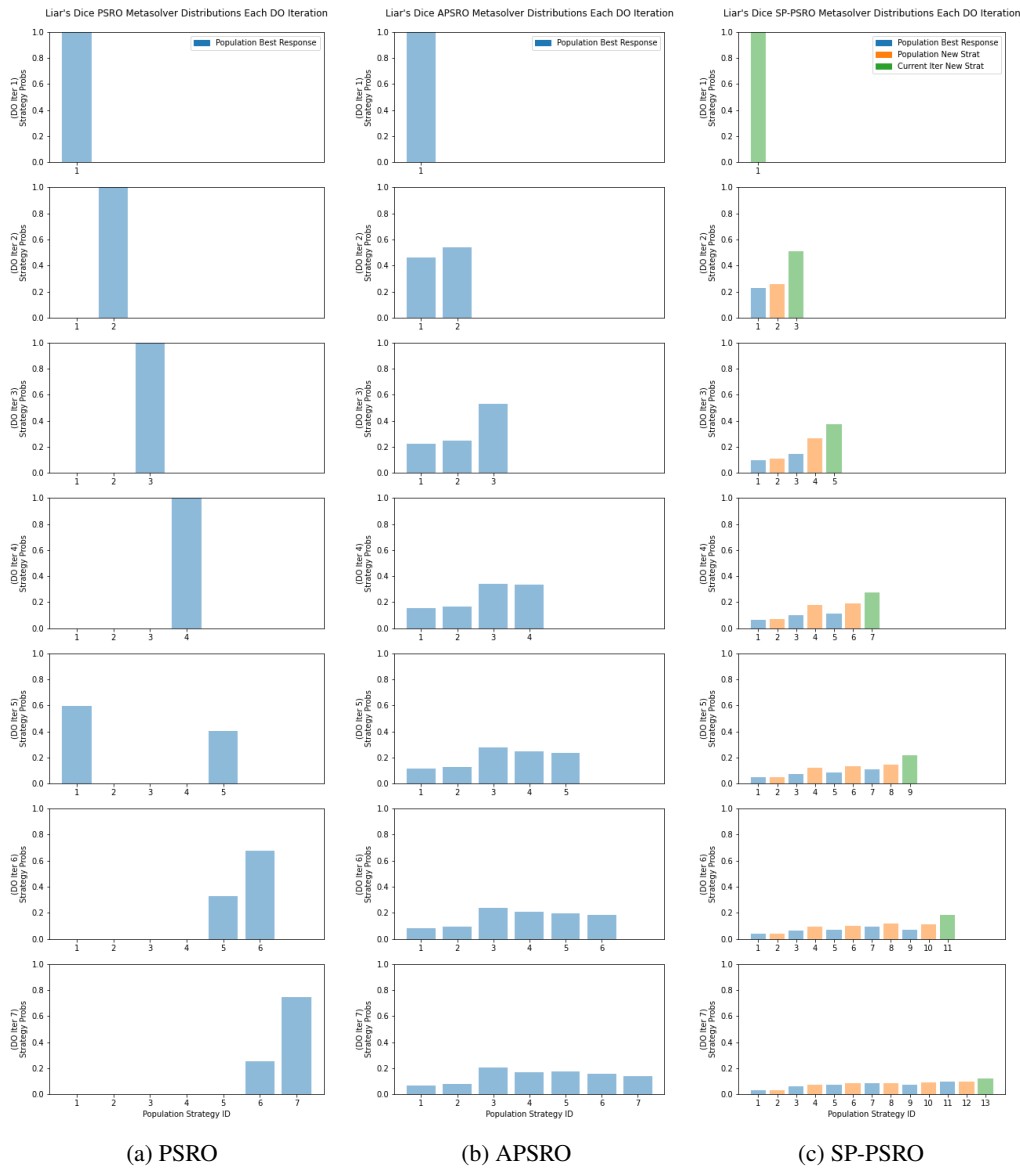

(a) PSRO        (b) APSRO        (c) SP-PSRO

Figure 16: PSRO, APSRO, and SP-PSRO mixed strategy solutions for Liar's Dice in the first 7 iterations of each algorithm.

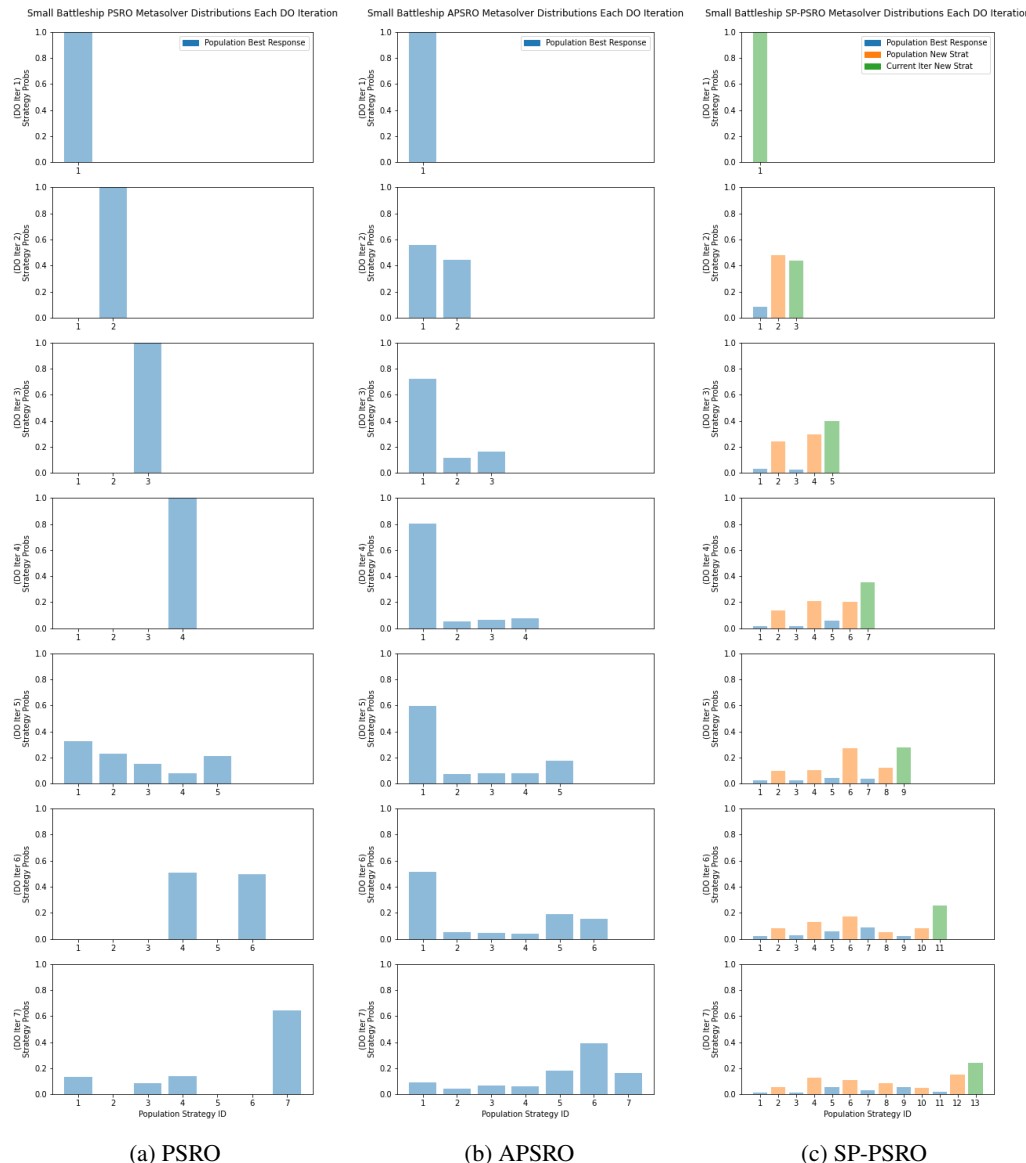

(a) PSRO        (b) APSRO        (c) SP-PSRO

Figure 17: PSRO, APSRO, and SP-PSRO mixed strategy solutions for Small Battleship in the first 7 iterations of each algorithm.

The NE to the restricted games, provided by PSRO in figures 16a, 17a, and 18a strongly favor the latest-added best responses in early iterations of the algorithm. While the latest best response is likely to be a strong strategy in a restricted game, an NE to the restricted game in early iterations of PSRO may not be the least exploitable mixed strategy against strategies outside of the population. APSRO, shown in figures 16b, 17b, and 18b optimizes for a no-regret solution to each restricted game where the opponent can use any strategy in the full game. By reducing exploitability against any possible opponent strategy rather than just those in $\Pi$, APSRO learns a notably distinct mixed strategy $\pi^r$ compared to PSRO that's safer for early-stopping.

SP-PSRO, in figures 16c, 17c, and 18c, also optimizes for a no-regret restricted-game mixed strategy against an unrestricted opponent. Like APSRO, SP-PSRO adds a best response $\beta_i$ for each player to the population in each iteration. In each iteration, SP-PSRO also adds the time-averaged new strategy $\bar{\nu}_i$ to the population. Unlike the best responses, $\bar{\nu}_i$ is included in the restricted distribution $\pi_i^r$ for the same iteration that it is trained. $\bar{\nu}_i$ is intended to very roughly approximate the strategy which most

reduces the exploitability of the population against an unrestricted opponent. In figures 16c, 17c, and 18c, we see that by their high weight in $\pi_0^r$ that the $\bar{\nu}_0$ we learn are indeed strong against unrestricted opponents. In each iteration, the no-regret mixed strategy $\pi_0^r$ puts the most weight on the "Current Iter New Strat" $\bar{\nu}_0$ strategy that was trained in that iteration. The "Population New Strat" strategies $\bar{\nu}_0$ are also assigned equal or higher weight in $\pi_0^r$ than the "Population Best Response" strategies $\beta_0$, showing that the time-averaged new strategies still serve as high-quality population strategies in iterations after those in which they are trained.

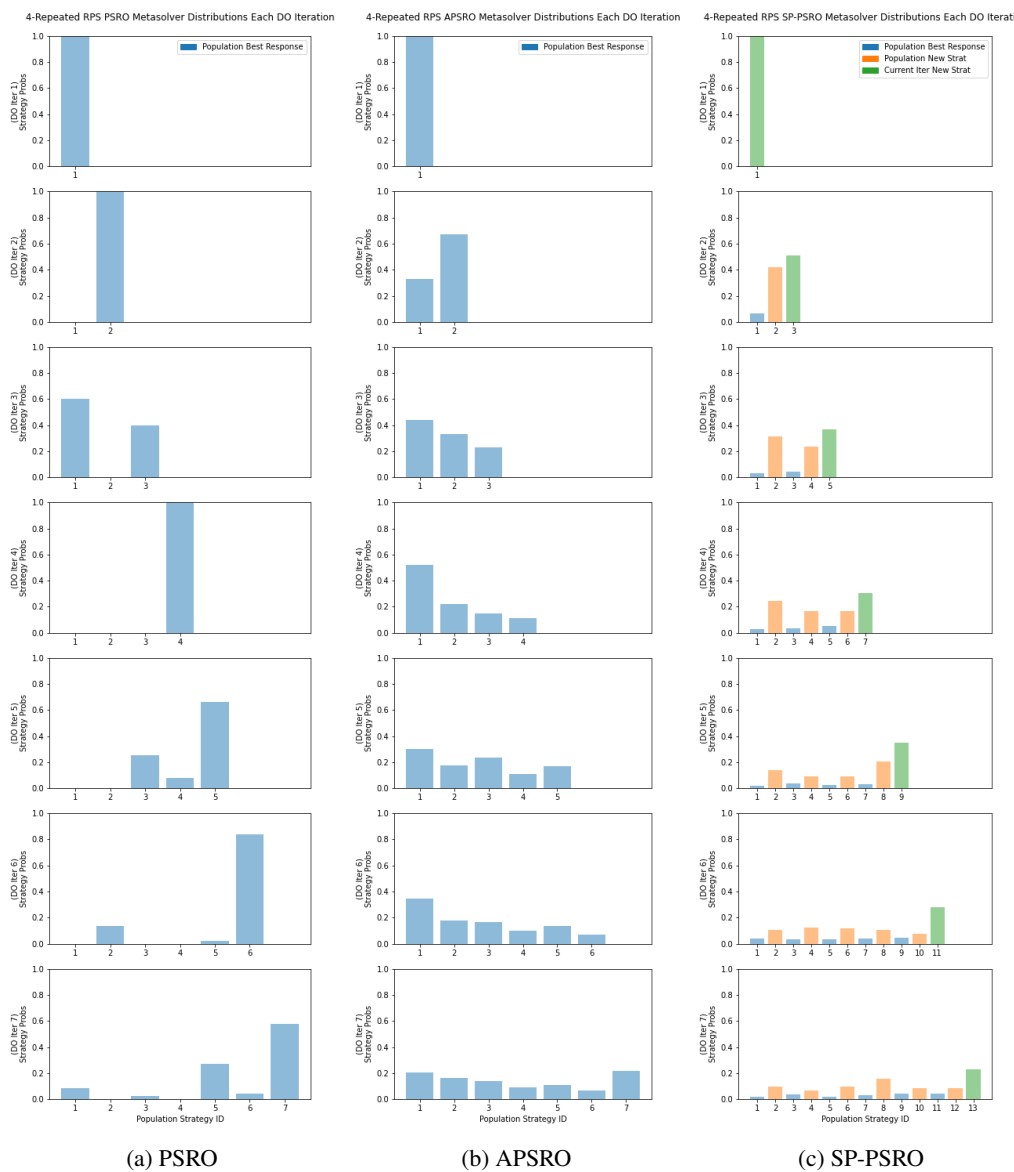

Figure 18: PSRO, APSRO, and SP-PSRO mixed strategy solutions for 4-Repeated RPS in the first 6 iterations of each algorithm.

## J  EXTENSIVE-FORM GAME ENVIRONMENTS

All extensive-form games tested with are from the OpenSpiel framework (Lanctot et al., 2019), and can be loaded using OpenSpiel with the following parameters:

**Leduc**  Game Name: `leduc_poker`
    Parameters: `{"players": 2}`

**Goofspiel**  Game Name: `goofspiel`
    Parameters: `{"imp_info": True, "num_cards": 5,`
    `"points_order":"descending",}`

**Tiny Battleship**  Game Name: `battleship`
    Parameters: `{"board_width": 2, "board_height":  2,`
    `"ship_sizes":  '[1]', "ship_values": '[1]',`
    `"num_shots": 2, "allow_repeated_shots": False,}`

**Small Battleship**  Game Name: `battleship`
    Parameters: `{"board_width": 2, "board_height": 2,`
    `"ship_sizes": "[1;2]", "ship_values": "[1;2]",`
    `"num_shots": 4, "allow_repeated_shots": False}`

**4x Repeated RPS**  Game Name: `repeated_game`
    Parameters: `{"num_repetitions": 4, "enable_infostate": True,`
    `"stage_game": "matrix_rps"}`

**Liar's Dice**  Game Name: `liars_dice`
    Parameters: `None`

Goofspiel and Repeated RPS are converted from simultaneous-move games into turn-based games using OpenSpiel's `convert_to_turn_based()` game transform. Repeated RPS is created from the `matrix_rps` game using the `create_repeated_game` game transform.

## K  TABULAR TRAINING DETAILS

For PSRO, APSRO, and SP-PSRO experiments with tabular Q-learning BRs, we used OpenSpiel's Python implementation of a tabular epsilon-greedy Q-learner. For all games, Q-learning hyperparameters were OpenSpiel defaults and were constant between PSRO, APSRO, and SP-PSRO: step size $= 0.1$ and epsilon $= 0.2$.

In all tabular experiments, when calculating the payoff for a strategy profile ($v_i(\pi)$), the payoff is calculated exactly using a full tree traversal.

### K.1  PSRO

In each iteration of tabular PSRO experiments, we first compute the restricted game payoff matrix, and we then use linear programming to find the NE of the restricted game. We train Q-learning agents for each player against the other's restricted game NE and add these to the population.

### K.2  SP-PSRO AND APSRO

We performed a hyperparameter sweep to find the number of episodes per iteration and number of Exp3 updates per iteration which minimize exploitability in Leduc poker after 35 iterations of SP-PSRO (Table 1). We used these hyperparameters for SP-PSRO and APSRO for tabular experiments in all games.

Each APSRO/SP-PSRO iteration, we set $\gamma$ to $\min\left(1, \frac{\sqrt{k \log k}}{(e-1)g}\right)$ where $g$ is an estimated upper bound on the total cumulative regret upon completion of the algorithm. In each iteration, we split the Q-learning training and Exp3 updates into 600 equally-sized batches each and alternate between a batch of Exp3 updates and a batch of Q-learning episodes until the end of the iteration. We target a total of $800,000$ episodes and $20,000$ Exp3 updates per iteration, and we repeat the following 600 times: for each player, perform $\lfloor 800,000/600 \rfloor = 1333$ Q-learning episodes followed by $\lfloor 20,000/600 \rfloor = 33$

Exp3 updates. While we target $800,000$ episodes and $20,000$ Exp3 updates per iteration, due to rounding, the actual amounts performed are smaller:

| | |
|---|---|
| episodes per iteration | 799,800 |
| Exp3 updates per iteration | 19,800 |
| Q-learning learning rate | 0.025 |
| Q-learning exploration $\epsilon$ | Constant, 0.2 |

Table 1: Tabular experiment details

For each Q-learning episode: we sample one policy $\pi_i$ for player $i$ from the distribution $\pi_i^r$ and then the sampled policy $\pi_i$ and the opponent Q-learning agent for $\beta_{-i}$ play an episode against each other. If the sampled policy $\pi_i$ corresponds to the new strategy $\nu_i$, it plays with $\epsilon$-greedy exploration. The opponent Q-learning agent always plays with $\epsilon$-greedy exploration. The episode is used to update the Q-learning agent for $\beta_{-i}$. The episode is also used to update the Q-learning agent for $\nu_i$, regardless of whether or not the chosen $\pi_i$ is $\nu_i$.

For tabular APSRO, we use the same code as for SP-PSRO, with the difference being that we do not create a new policy $\nu_i$.

## L    DEEP RL TRAINING DETAILS

For deep RL experiments, we use the same RL best response hyperparameters in PSRO, APSRO, and SP-PSRO. When RL best responses are calculated, an independent best response learning process is performed for each player. RL hyperparameters for each game were selected based on sample efficiency and final performance against a fixed opponent.

Our deep RL code was built on top of the RLlib framework (Liang et al., 2018b), and any hyperparameters not specified are the version 1.0.1 defaults.

### L.1    PSRO

The PSRO restricted payoff matrix is estimated using 3000 evaluation rollouts per policy matchup, and the meta-game NE is calculated using 2000 iterations of Fictitious Play (Brown, 1951). APSRO and SP-PSRO skip calculating the restricted payoff matrix. We do not count experience used to generate payoff matrix utilities in comparisons with PSRO.

### L.2    APSRO

In deep RL experiments, we use the Multiplicative Weights Update (MWU) algorithm Cesa-Bianchi & Lugosi (2006); Freund & Schapire (1999) as our no-regret solver for APSRO with a learning rate of 0.1, updating every 10th RL iteration. Action payoffs for MWU corresponding to expected utilities for population policies in $\Pi_i^t$ against the current $\beta_{-i}$ are estimated by averaging the empirical payoffs from the last 1000 rollouts in which each population policy was sampled. Exploitability is measured against the time-average of the MWU mixed-strategy from each APSRO iteration.

### L.3    SP-PSRO

For SP-PSRO, we use the same MWU no-regret solver and parameters as we do with APSRO, where the actively-learning new strategy $\nu_i$ is included as an action for the no-regret solver. Because $\nu_i$ would by default only collect experience when the no-regret solver samples it, we additionally provide $\nu_i$ with off-policy experience from all other policies in the population $\Pi_i^t$ when they are sampled and generate experience as well.

We train the time-average $\bar{\nu}_i$ of $\nu_i$ as a neural network, and to do so, we save all experience generated by $\nu_i$ to a buffer using reservoir sampling (Vitter, 1985; Heinrich & Silver, 2016) with a maximum capacity of 2e6 samples. After BR training is complete, we use supervised learning to train a softmax policy on the reservoir buffer data with cross-entropy loss on actions given observations to distill the

time-average of the new policy $\nu_i$. To ensure that enough experience from $\nu_i$ is always generated and added to the reservoir buffer, a small fixed portion $p$ of all experience rollouts in the BR training process is forced to be played as a matchup between $\nu_i$ and a non-exploring evaluation copy of $\beta_{-i}$.

For Liars Dice, $p = 0.05$, and for Small Battleship and 4x Repeated RPS, $p = 0.1$. We train each $\bar{\nu}_i$ on the reservoir buffer data with a learning rate of 0.1 for 10,000 SGD batches. We use an MLP with three 128-unit layers and ReLu activations for $\bar{\nu}_i$ in all games.

Exploitability is measured against the time-average of the MWU mixed-strategy from each SP-PSRO iteration where $\bar{\nu}_i$ is used to represent the new strategy.

When the new population strategy $\nu_i$ and the BR $\beta_{-i}$ collect experience against each other, unless otherwise stated, they both use and play against exploring $\epsilon$-greedy versions of each other.

### L.4 BEST RESPONSES

Hyperparameters to train deep RL best responses for each game are provided below. We use DDQN (Van Hasselt et al., 2016) to train RL best responses for all deep RL experiments. Any hyperparameters not listed are default values in RLlib (Liang et al., 2018a) version 1.0.1.

| | |
|---|---|
| algorithm | DDQN |
| circular replay buffer size | 50,000 |
| prioritized experience replay | No |
| total rollout experience gathered each iter | 2048 steps |
| learning rate | 0.0026 |
| batch size | 4096 |
| optimizer | Adam (Kingma & Ba, 2014) |
| TD-error loss type | MSE |
| target network update frequency | every iteration |
| MLP layer sizes | [128, 128] |
| activation function | ReLu |
| discount factor $\gamma$ | 1.0 |
| best response RL process stopping condition | 7.5e5 timesteps |
| exploration $\epsilon$ | Linearly annealed from 0.06 to 0.001 over 2e5 timesteps |

Table 2: Liar's Dice Deep RL Best Response Hyperparameters

| | |
|---|---|
| algorithm | DDQN |
| circular replay buffer size | 200,000 |
| prioritized experience replay | No |
| total rollout experience gathered each iter | 1024 steps |
| learning rate | 0.0019 |
| batch size | 2048 |
| optimizer | Adam (Kingma & Ba, 2014) |
| TD-error loss type | MSE |
| target network update frequency | every 1e5 timesteps |
| MLP layer sizes | [128, 128, 128] |
| activation function | ReLu |
| discount factor $\gamma$ | 1.0 |
| best response RL process stopping condition | 3e5 timesteps (Repeated RPS) & 7.5e5 timesteps (Battleship) |
| exploration $\epsilon$ | Linearly annealed from 0.06 to 0.001 over 2e6 timesteps |

Table 3: 4x Repeated RPS and Small Battleship Deep RL Best Response Hyperparameters

## M  COMPUTATIONAL COSTS

Experiments were run on local machine with 128 logical CPU cores, 4 Nvidia RTX 3090 GPUs, and 512GB of RAM. Each tabular experiment run used a single core, and each deep RL experiment run used up to 5 CPU cores per player to train best responses and up to 4 CPU cores to evaluate meta-game empirical payoffs, for a maximum total of 14 cores per deep RL experiment. All deep RL experiments individually used less than 5GB of VRAM. Tabular and deep RL experiments had durations between 1 and 7 days.

Concerning storage requirements, although PSRO-based methods need to store network weights for generated population policies on disk, we found disk-space usage to be a non-issue as population sizes were generally on the order of at most a hundred policies. Table 4, describes the final population size and disk usage for each game tested in deep RL experiments. SP-PSRO generates twice as many policies as PSRO and APSRO because SP-PSRO also saves and adds each iteration's average new strategy to the population.

|  | Final Population Size | Disk Usage Per Policy | Total Disk Usage |
|---|---|---|---|
| **PSRO** | 32, 26, 74 | 152KB, 152KB, 148KB | 4.864MB, 3.952MB, 10.952MB |
| **APSRO** | 32, 26, 74 | 152KB, 152KB, 148KB | 4.864MB, 3.952MB, 10.952MB |
| **SP-PSRO** | 64, 52, 148 | 152KB, 152KB, 148KB | 9.728MB, 7.904MB, 21.904MB |

Table 4: Population sizes and policy disk-space usage for Liar's Dice, Small Battleship, and 4-Repeated RPS

## N  CODE

Code for deep RL experiments is available at `https://github.com/indylab/sp-psro` under the MIT license. Our code is built on top of the OpenSpiel (Lanctot et al., 2019) and RLlib (Liang et al., 2018a) frameworks, both of which are open source and available under the Apache-2.0 license.

