# OpenReview forum: "Toward Optimal Policy Population Growth in Two-Player Zero-Sum Games"
_ICLR.cc/2024/Conference — ICLR 2024 poster_

### Official Review · Reviewer_EJCG · 2023-10-16

**Soundness:** 3 good
**Presentation:** 3 good
**Contribution:** 2 fair
**Rating:** 6
**Confidence:** 4

**Summary:**

This paper considers the redesign of Policy-Space Response Oracles (PSRO) as an _anytime algorithm_---designed so that it can stop at any time and return the best result found so far, for two-player zero-sum games. The authors begin by extending the Double Oracle (DO), the precursor of PSRO, to the anytime setting. They accomplish this through their proposed algorithm Anytime Double Oracle (ADO) that considers differing restricted games per-player that leave the opponent always unrestricted. They show theoretically that ADO inherits the limit convergence properties of DO. This is followed by their introduction of Anytime PSRO (APSRO), an algorithm that "never increases exploitability by much" when compared to PSRO. APSRO works by continually computing the estimated solution using a regret minimization algorithm. Finally, they introduce Self-Player PSRO (SP-PSRO) that modifies APSRO to add the BR(BR) at each iteration instead of just a BR. SP-PSRO and APSRO are evaluated against PSRO on a suite of small two-player zero-sum games.

**Strengths:**

- The anytime property in game-solving algorithms is important as more heuristics for strategy exploration are being considered. This is particularly true because adding best responses at each iteration is a heuristic method, resulting in the increasing exploitability sometimes observed in PSRO.
- Visualizations of didactic examples of the algorithm are very welcome and clear.
- Evaluated on a diversity of games --- albeit, all small games presumably to compute exact exploitability.

**Weaknesses:**

- The premise of the paper, PSRO/DO can result in increases in exploitability between iterations, is not sufficiently justified or explained. Wouldn't a more straightforward approach to just be take the best-found-so-far solution across PSRO iterations? With this slight modification on the return statement, you obtain the desired anytime property.
- Another premise of the paper is that "PSRO and APSRO add pure-strategy (i.e. deterministic) best responses in each iteration." PSRO _does not_ have this requirement, and in fact, begins by adding a uniformly stochastic policy to the strategy set. As it is relatively common for implementations of PSRO to include only deterministic policies, I am fine with this being used as a premise, but I think this needs to be directly discussed in the text. This fact also has side-effect of suggesting an alternative simpler approach to APSRO/SP-PSRO in differing methods of computing stochastic policies.
- ADO is marginally novel compared to Range of Skill (RoS) and DO, and I think its limitations are not discussed well enough. To me, the main intuition behind DO/PSRO is when you cannot consider analytical solutions to game reasoning, and instead need to consider a restricted/empirical version of the game. ADO considers a partially restricted version of the game (only expanding a single players policy-space), and as a result is intractable and bears common complaints to direct game reasoning. The authors should discuss this and the complexity trade-offs more directly.
- The authors should address the focus on NE, when PSRO is flexible to any chosen solution concept. They should also state for all of their algorithms if they are constraining this aspect of PSRO.
- Very limited analysis of the design of APSRO/SP-PSRO. Consider including ablation-style experiments.
- Restricted games and empirical games are used interchangeably when they are not interchangeable objects.


- This is more minor, but I think is very important: the language and notation are more posed to a game theory audience than to the ML audience of ICLR. Following the notation and definitions in say the PSRO paper for example would make this paper significantly more approachable by this community. I personally find trouble whenever the RL definitions/notations are conflated with the notations used within the game-theoretic reasoning. For example, when you see \pi, is it a single policy, or a distribution over policies (i.e., solution)?

**Questions:**

- "In addition to introducing APSRO, we also build on APSRO by adding to the population in each iteration an approximation of the myopically optimal policy, that is, the stochastic policy that maximally lowers the exploitability of the least-exploitable distribution of the resulting population."
  - If I understand this claim correctly, this is not a new insight, and the work discussed in A.2 presented this idea?
- I would suggest the authors consider a different name than Self-Play PSRO. This makes me think of PSRO with the MSS set to Self-Play, and doesn't really hint at all as to the nature of the underlying algorithm.
- Why bother forcing SP-PSRO to train off policy?
  - The focus of this work is on anytime stopping not on computational efficiency.
  - Have the authors tried using on-policy algorithms? This could result in further benefits.
- "Moreover, adding mixed strategies can generally reduce exploitability faster than adding pure strategies."
	- Is another way to make this argument that adding mixed strategies enables us to add multiple pure-strategies per iteration; therefore, we can generally solve the game quicker---by the measure of number of iterations?
	- This seems to suggest a degenerate solution of just immediately solving the full game, which of course we can't generally do.
- "Why the episode is terminated, the time average of v is added to the population." Why is episode used here? Do you only train your policy for a singular episode?
- Limitations should discuss that while SP-PSRO uses same budget as APSRO, they're both way more expensive than PSRO.
- Limitations need to also discuss that this analysis is on NE, PSRO is more general


Exp 6.1
- You've had to add lambda to apply your algorithms to these games. What is the impact of lambda on the performances you've demonstrated?
- "As shown in Figure 4, unlike PSRO, APSRO tends not to increase exploitability" I don't think eyeballing Fig 4 or 5 really provides enough evidence to support this claim.
	- PSRO only looks guilty of a large exploitability increase in Leduc Poker, and all other both algorithms are erratic.
- "Note that APSRO and SP-PSRO only reach an ϵ-NE because they use a finite number of regret minimization updates to determine the restricted distribution, while PSRO is able to exactly compute a NE."
	- I would have preferred the authors to have included an analysis of this.
	- A/SP-PSRO with exact equilibrium computation.
	- PSRO with regret minimization for computing equilibrium.
	- Would help better understand where the benefits are coming from.

Exp 6.2
- "We hypothesize that this is due to the APSRO iterations not being long enough for the no-regret process to converge."
	- Thank you for including this point.
	- I was wondering if you had any insights into how we could know when we should apply APSRO based on this?

---

> ### Author Response · Authors · 2023-11-21
> **Author Response**
>
> # Introduction
>
> We appreciate the thorough review and insightful comments provided by Reviewer EJCG. Your feedback has been instrumental in refining our paper. We have addressed each of your points and have changed the paper to reflect these changes. You can see the changes in blue in the new text.
>
> ## 1. Outputting the Best Found So Far Solution
>
> **Reviewer's Comment:**
> The reviewer suggests outputting the best-found-so-far solution across PSRO iterations to obtain the desired anytime property.
>
> **Response:**
> This is a reasonable suggestion. We had thought about this before submitting the paper but had decided not to include it because it does not perform as well. We have included this in Appendix F.4 in the final version of the paper. It performs better than PSRO but slightly worse than APSRO. A similar result is also evident in Figure 4.a where PSRO performance is near monotonic. Furthermore, we want to point out that this best-so-far technique could also be applied to APSRO and SP-PSRO.
>
> ## 2. PSRO Pure Strategies
>
> **Reviewer's Comment:**
> The paper originally did not clarify that common implementations of PSRO add pure strategies in each iteration.
>
> **Response:**
> We've revised the text (highlighted in blue) to specify that *common* implementations of PSRO add pure strategies every iteration.
>
> ## 3. ADO and Its Novelty
>
> **Reviewer's Comment:**
> Concerns were raised about the novelty of ADO and its comparison with Range of Skill (RoS) and DO.
>
> **Response:**
> We are upfront in our paper about the similarities of ADO with RoS and DO. We do not view ADO as our primary contribution but rather as a theoretical foundation on which we build the other scalable algorithms. We added to the text: “ADO primarily serves as a foundational element for the development of our subsequent algorithm, APSRO. This foundational role is crucial as it lays the groundwork for APSRO's convergence guarantees.”
>
>
> ## 4. Focus on Nash Equilibrium (NE)
>
> **Reviewer's Comment:**
> The paper primarily focuses on NE, while PSRO is adaptable to various solution concepts.
>
> **Response:**
> We've added a line clarifying that while our current focus is on two-player zero-sum games and NE. These techniques (at least APSRO) directly carry over to the TMECor solution concept for team games [1].
>
> We are not sure whether that is the type of refinement that the reviewer is asking about. Or perhaps the reviewer is suggesting some normal-form refinement such as CCE. Or, perhaps the reviewers is suggesting some extensive-form refinement such as trembling hand. If the reviewer clarifies what sorts of refinements the reviewer is alluding to, we can try to provide further answers.
>
>
> ## 5. Ablations
>
> **Reviewer's Comment:**
> Consider including ablation-style experiments.
>
> **Response:**
> In the appendix, we've included additional experiments with SP-PSRO not-anytime and SP-PSRO last-iterate. We believe this covers the additions present in our algorithms. Are there any other particular ablations the reviewer would like us to include?
>
> ## 6. Clarifications and Language
>
> **Reviewer's Comment:**
> Last two weakness bullets: need for clearer language and notation, particularly for an ML audience.
>
> **Response:**
> We have revised the language for clarity, aligning it more closely with the ML audience's familiarity. Notations have been adjusted for consistency and comprehensibility. In particular, we have changed any mention of “empirical game” to “restricted game”. Changes are in blue.
>
>
> ## 7. Myopically Optimal Policy
>
> **Reviewer's Comment:**
> Clarification on the claim about adding an approximation of the myopically optimal policy. Also the question about adding mixed strategies.
>
> **Response:**
> The work discussed in A.2 does introduce the idea of a least-exploitable distribution, which we point out in the main text. However, that literature does not introduce the concept of adding the best mixed strategy to minimize exploitability the most.
>
> You are right that this line of text is confusing though. The best mixed strategy to add, as you point out (and we also point out in our paper), is the actual Nash equilibrium. This is of course too hard to compute up front so we add a mixed strategy that does something similar to fictitious play that happens to have good empirical performance. As a result, we have changed the line to the following: “In addition to introducing APSRO, we also build on APSRO by adding to the population in each iteration a stochastic policy that is trained via an off-policy procedure.”

---

> > ### Author Response · Authors · 2023-11-21
> > **Author Response Continued**
> >
> > ## 8. Self-Play PSRO Name
> >
> > **Reviewer's Comment:**
> > The name 'Self-Play PSRO' might be confusing.
> >
> > **Response:**
> > We acknowledge this potential confusion. The name was chosen to reflect the mixed strategy found via training against the opponent's best response. We are open to renaming it based on suggestions you may have.
> >
> >
> > ## 9. Training Off Policy in SP-PSRO
> >
> > **Reviewer's Comment:**
> > Query about the necessity of off-policy training in SP-PSRO.
> >
> > **Response:**
> > Sample efficiency is a significant concern for us. SP-PSRO is introduced with efficiency as the main motivation, which justifies the use of an off-policy algorithm to train the new strategy. Using an on-policy algorithm would add significant experience overhead in training the new strategy in combination with the standard APSRO/PSRO best response.
> >
> > It is also possible that we are not understanding correctly what the reviewer is asking. In that case, please clarify the question/comment and we will elaborate further. For example, is the reviewer talking about on-policy for the best responder or for the new strategy?
> >
> > ## 10. Adding mixed strategies
> >
> > **Reviewer’s Comment**
> > The reviewer questions if adding mixed strategies speeds up game-solving by effectively incorporating multiple pure strategies per iteration, while acknowledging the impracticality of immediately solving the entire game.
> >
> > **Response**
> > See our response to number 7.
> >
> >
> > ## 11. Question on Episode for Time Average
> >
> > **Reviewer’s Comment**
> > "Why the episode is terminated, the time average of v is added to the population." Why is episode used here? Do you only train your policy for a singular episode?
> >
> > **Response**
> > Thank you for noticing this. We have changed the wording to “When the *iteration* is finished, the time-average $\bar{\nu_i}$ of $\nu_i$ is added to the population.”
> >
> >
> > ## 12. Limitations and Budget
> >
> > **Reviewer's Comment:**
> > Limitations should discuss that while SP-PSRO uses same budget as APSRO, they're both way more expensive than PSRO.
> >
> > **Response:**
> > This is incorrect. APSRO uses the same budget as PSRO in terms of the number of episodes / environment interactions. Wall clock time is slightly slower. We have added the following to clear this up: “Importantly, compared to PSRO, APSRO uses the same amount of episodes and environment interactions. The only difference between the two is that APSRO changes the restricted distribution dynamically via a no-regret procedure.”
> >
> > ## 13. Limitations about Different Solution Concepts
> >
> > **Reviewer’s Comment**
> > Limitations need to also discuss that this analysis is on NE, PSRO is more general
> >
> > **Response**
> > We have added to the limitations that our method only focuses on Nash equilibrium. See also our response to number 4.

---

> > > ### Author Response · Authors · 2023-11-21
> > > **Author Response Continued**
> > >
> > > ## 14. Experiment 6.1
> > >
> > > **Reviewer’s Comment**
> > > What is the impact of lambda on the performances you've demonstrated?
> > >
> > > **Response**
> > > We simulate best response training dynamics in these normal form experiments to demonstrate the relative performance of our algorithms on games that are quick and easy to run. So the lambda term should not be viewed as a hyperparameter that needs to be tuned, but rather a fundamental part of the game and environment experimental setup. Since you have asked for ablations on lambda, we have included them in the appendix. These results show that changing the lambda parameter does not change the performance that much.
> > >
> > > **Reviewer’s Comment**
> > > As shown in Figure 4, unlike PSRO, APSRO tends not to increase exploitability
> > >
> > > **Response**
> > > We have removed this line.
> > >
> > > **Reviewer’s Comment**
> > > I would have preferred the authors to have included an analysis of A/SP-PSRO with exact equilibrium computation and PSRO with regret minimization for computing equilibrium.
> > >
> > > **Response**
> > >
> > > *A/SP-PSRO with exact equilibrium computation.*
> > > The exact version of APSRO is ADO (Algorithm 1). We present experimental results for ADO in Appendix C. SP-PSRO has no exact counterpart because it relies on computing a time average of the new strategy over the course of a finite, boundedly accurate regret-minimization procedure.
> > >
> > > *PSRO with regret minimization for computing equilibrium.*
> > > The NE for the restricted game solved by DO and PSRO is easily tractable in most realistic scenarios, and we use exact methods rather than regret-minimization to solve DO’s/PSRO’s restricted game. PSRO with regret minimization for computing equilibrium to this restricted game will provide an equally or slightly less accurate restricted solution than an exact computation, depending on how long the regret minimization is run.
> > >
> > >
> > > ## 15. Experiment 6.2
> > >
> > > **Reviewer's Comment:**
> > > Concerns about the applicability of APSRO in different scenarios.
> > >
> > > **Response:**
> > > We acknowledge the limitation in predicting when APSRO might outperform PSRO. This mirrors the inherent uncertainties in PSRO performance. We have, however, observed consistent improvements in most games tested.
> > >
> > >
> > > # Conclusion
> > >
> > > We have made all the changes in blue for easy reference. We believe these amendments address your concerns effectively. Please let us know if there are any further modifications you would recommend.
> > >
> > > Thank you for your valuable feedback, which has greatly contributed to enhancing the quality of our work.
> > >
> > > [1] McAleer, Stephen, et al. "Team-PSRO for Learning Approximate TMECor in Large Team Games via Cooperative Reinforcement Learning." Thirty-seventh Conference on Neural Information Processing Systems. 2023.

---

> > > > ### Comment · Reviewer_EJCG · 2023-11-22
> > > >
> > > > Thank you for addressing my questions and comments. I have increased my score correspondingly. My largest remaining reservation is that the primary hypothesis of the paper seems like it should be different. Upon my initial reading I assumed the through-line was just focusing on obtaining the _anytime_ property within PSRO. As the authors confirmed with me, adding the anytime quality in PSRO is actually quite straightforward to achieve. Another point of interest in this paper is to reduce the computational cost of learning-based game-solving (similar to methods Mixed-Opponents, and Dyna-PSRO) through new heuristic strategy exploration methods.
> > > >
> > > > After reading through all of the reviews and rebuttals, I would conclude now that the paper is primarily focused in developing strategy exploration methods that are _anytime_. In other words, how can I _add_ a new policy that is unlikely to hurt the empirical game. I realize that this may seem very pedantic, but I think this is framing is key to understanding where this work sits in the literature. First, it removes the degenerate solution I previously mentioned (return best-found-so-far to achieve anytime). Second, it elevates the importance of considering where this work contributes to strategy exploration (how to choose which policy to add to an empirical/restricted game) specifically. I'll now unpack that claim a bit more.
> > > >
> > > > Strategy exploration, much like standard exploration, may choose to take short-term penalties for long-term games. In PSRO, this amounts to adding potentially poor policies that are useful in generating more performant policies later into game solving. Now, returning to _anytime_-ness, I believe this paper is trying to say we'd like to add exploration policies, but only ones that are also "good". I think this is where my main reservations lie. It's not clear based on this paper that adding these "good" policies are good in the long-term, but may be important if the algorithm may terminate in the short term. So, to me, the motivation of _anytime_-ness feels not fully justified. Furthermore, we also tend to have some insight as to how many iterations will occur, because these algorithms take so long to run.
> > > >
> > > > On the other hand, it appears this paper has contributed a performant strategy exploration method. As the included experiments demonstrate strong empirical performance. I think this paper overall would've been much stronger pitched from this angle, with qualities like anytime being a minor commentary.
> > > >
> > > > In total, I think this paper has value and does things well (including algorithm, didactic examples, lots of experiments and ablations). I just don't buy the narrative of the paper.
> > > >
> > > > I realize the discussion period is limited at this point. But I would love the author's thoughts on what I've outlined here. The paper is currently borderline, and would love to be convinced to make the decision clear.
> > > >
> > > >
> > > > > 1. Best Found So Far
> > > >
> > > > Thank you for including this ablation. I think it really helps frame the "anytime" problem from a very straightforward baseline.
> > > >
> > > >
> > > > > 4. Focus on Nash Equilibrium
> > > >
> > > > Adding that the focus is on two-player zero-sum games and NE addresses my concern. My original point is simply that PSRO can be used to solve for arbitrary solution concepts, and that it's important to acknowledge you're viewing a subclass of PSRO algorithms (solving for _specific_ solution concepts).
> > > >
> > > >
> > > > > 9. Training Off Policy
> > > >
> > > > I acknowledge that cost is very limiting in learning-based game-solving algorithms. My point on this was meant to reflect that this seems to be an orthogonal hypothesis to the primary purpose of the paper: _anytime_ game-solving algorithms.

---

> ### Author Response · Authors · 2023-11-22
> **Author Response**
>
> Thank you for your thoughtful and detailed feedback. We appreciate your reconsideration of the paper's score.
>
> The main focus of our paper is to introduce new PSRO methods that achieve good performance even before convergence. In large games such as Starcraft, Dark Chess, and Stratego, PSRO may never converge because it requires so many policies. As a result, it is crucial to achieve good performance even before convergence.
>
> Our method, SP-PSRO, is designed with this practicality in mind, ensuring that the strategies developed in the initial phases not only effectively explore the strategy space but also greatly improve performance before convergence. This approach aligns with the anytime-ness concept, which we believe is not merely a theoretical aspect but a critical feature in the real-world application of our algorithm. It ensures that at any point of early termination, the strategies are optimally effective, addressing the challenge of computational feasibility in large games.
>
> We have added the following sentences to the introduction to better reflect this framing:
>
> - "In large games like dark chess and Starcraft, where PSRO may never converge, the early performance holds paramount importance. Our approach with SP-PSRO is tailored to this reality, ensuring robust performance from the outset. Recognizing that the completion of the full training procedure in such extensive games is a rare occurrence, the anytime property of our proposed method takes on a critical role, delivering viable strategies at any stage of the iterative process."
>
> - "Our empirical results demonstrate SP-PSRO's superior performance in reducing exploitability before convergence across various games, a testament to its practical effectiveness."
>
> - "While APSRO serves as a foundational concept in our research, the leap to SP-PSRO marks a significant advancement, particularly in terms of reducing exploitability before PSRO has neared convergence."
>
> We would also like to make the clarification that anytime-ness is achieved by solving a modified restricted game with a solution resilient to adding strategies that would normally hurt standard PSRO in the short term. Building on the benefits of anytime-ness, better strategy exploration helps us achieve lower exploitability early on in training. This is done by more sample-efficiently building effective populations able to represent important interactions in the full game. By providing both anytime-ness and better strategy exploration, the methods introduced in the work help us achieve reliably better performance in earlier stages of training and under smaller experience budgets.
>
> We hope this clarification strengthens the understanding of our work's contributions and its practical significance in the field. Please let us know if there is anything we can further change.

---

### Official Review · Reviewer_C3x7 · 2023-10-31

**Soundness:** 3 good
**Presentation:** 3 good
**Contribution:** 3 good
**Rating:** 6
**Confidence:** 2

**Summary:**

I have read the author's rebuttal, and I have decided to keep my score unchanged. The main reason is the resource consumption issue of PSRO-like methods in large-scale problems. The author mentioned Barrage Stratego, which is still too small for me (as a RL researcher). I acknowledge that large-scale problems, multi-player games, and multi-agent issues often do not guarantee convergence to Nash equilibrium, but the vast majority of problems we actually face are of this kind. Therefore, I encourage the author to try PSRO-like methods in these problems to expand the impact of the article.

========================================================================================================

This paper addresses the issue of increased exploitability in deep reinforcement learning methods, such as Policy Space Response Oracles (PSRO), in competitive two-agent environments. The authors propose anytime double oracle (ADO), an algorithm that ensures exploitability does not increase between iterations, and its approximate extensive-form version, anytime PSRO (APSRO). Furthermore, the paper introduces Self-Play PSRO (SP-PSRO), which incorporates an approximately optimal stochastic policy into the population in each iteration. Experiments demonstrate that APSRO and SP-PSRO have lower exploitability and near-monotonic exploitability reduction in games like Leduc poker and Liar's Dice, with SP-PSRO converging much faster than APSRO and PSRO.

**Strengths:**

Originality: The paper presents novel algorithms (ADO, APSRO, and SP-PSRO) that address the exploitability issue in competitive two-agent environments, which is a significant contribution to the field of deep reinforcement learning.
Quality: The proposed algorithms are well-motivated, theoretically grounded, and empirically validated through experiments on various games.
Clarity: The paper is well-written, with clear explanations of the algorithms and their motivations, making it easy to understand the authors' contributions.
Significance: The proposed algorithms have the potential to significantly improve the performance of deep reinforcement learning methods in large games, making them more applicable to real-world problems.

**Weaknesses:**

Scalability: While the proposed algorithms show promising results in the tested games, it is unclear how they would scale to even larger games or more complex environments, such as limit Texas Hold'em or Atari games. The paper could provide more insights into the scalability and potential limitations of the proposed algorithms in these settings.

Comparison: The paper could benefit from a more comprehensive comparison with other state-of-the-art methods, including non-PSRO methods such as population-based training and league training (e.g., AlphaStar). This would provide a clearer understanding of the relative performance of the proposed algorithms and their advantages over existing approaches.

Generalization: The paper focuses on competitive two-agent environments, but it would be interesting to see how the algorithms could be adapted or extended to handle multi-agent or cooperative settings, where exploitability is harder to compute but more general problems are commonly encountered.

**Questions:**

Similiar to the weaknesses:

Can the authors provide more insights into the scalability of the proposed algorithms, particularly in the context of larger games or more complex environments, such as limit Texas Hold'em or Atari games?

How do the proposed algorithms compare to other state-of-the-art methods, including non-PSRO methods such as population-based training and league training (e.g., AlphaStar), in terms of exploitability reduction and convergence speed?

Are there any plans to extend the proposed algorithms to multi-agent or cooperative settings? If so, what challenges do the authors foresee in adapting the algorithms to these settings, where exploitability is harder to compute but more general problems are commonly encountered?

---

> ### Author Response · Authors · 2023-11-21
> **Author Response**
>
> ## Introduction
> We appreciate the thoughtful review of our paper and your vote of acceptance. However, we believe that the concerns raised can be addressed in future work and that our current contributions stand on their own.
>
> ## Scalability
> Our algorithms are not inherently limited by the size of the game. The theoretical underpinnings of ADO, APSRO, and SP-PSRO ensure scalability. Their performance in larger games is a matter of computational resources rather than algorithmic capability. Indeed, PSRO has already been shown to scale to large games such as Barrage Stratego. Our experiments on Leduc poker and Liar's Dice, while seemingly limited, were chosen due to their relevance in the field. These games, though not as large as limit Texas Hold'em or Atari games, still present significant challenges in terms of state space and strategy complexity, providing a robust testbed for our algorithms. Further demonstrating performance on large games is outside the scope of this paper.
>
> ## Comparison with Other Methods
> It is crucial to understand that our algorithms are specifically designed to converge to Nash equilibrium. While comparisons with population-based and league training methods could be interesting, these methods are not guaranteed to converge to a Nash equilibrium, as shown in previous works on PSRO [1]. We did not include these other methods such as PSRO Rectified Nash PSRO [3] because it has already been shown not to converge to a Nash equilibrium. However, since the reviewer asked, we have included results with PSRO Rectified Nash in the appendix.
>
> ## Generalization
> Great point. We agree that extending our approach to multi-agent settings is a valuable avenue. While our current work focuses on two-agent environments, the methodologies and insights gained provide a stepping stone for tackling more complex scenarios. Adapting our algorithms for multi-agent settings is part of our ongoing research. We are exploring ways to compute exploitability and develop strategies in these environments. One particular path we are exploring is to combine techniques from this paper with techniques from this recent paper [2] for finding TMECor in team games. Thankfully, TMECor provides a similar notion of exploitability as in two-player zero-sum games. We have added text in the discussion and limitation about this aspect.
>
> ## Conclusion
> Our work represents a significant step forward in improving scalable PSRO algorithms while maintaining convergence guarantees. While there are always avenues for further research and improvement, we believe our contributions are solid, well-founded, and a valuable addition to the field. We hope this rebuttal clarifies our position and the strengths of our work. Thank you for your consideration.
>
> [1] McAleer, Stephen, et al. "Pipeline PSRO: A scalable approach for finding approximate nash equilibria in large games." Advances in neural information processing systems 33 (2020): 20238-20248.
>
> [2] McAleer, Stephen, et al. "Team-PSRO for Learning Approximate TMECor in Large Team Games via Cooperative Reinforcement Learning." Thirty-seventh Conference on Neural Information Processing Systems. 2023.
>
> [3] Balduzzi, David, et al. "Open-ended learning in symmetric zero-sum games." International Conference on Machine Learning. PMLR, 2019.

---

### Official Review · Reviewer_xLd8 · 2023-11-04

**Soundness:** 3 good
**Presentation:** 2 fair
**Contribution:** 3 good
**Rating:** 6
**Confidence:** 3

**Summary:**

The authors consider the problem of approximating a Nash equilibrium in a two-player zeros-sum imperfect information games with perfect recall. They propose a new double oracle algorithm, the anytime double oracle (ADO) algorithm that builds a population of policies by adding the best response against an optimal policy of restricted games (restricted by using only a mixture of policies present in the population). They show that the exploitability of ADO, contrary to the one of DO, decrease at each iteration. They also propose the APSRO algorithm an approximation of ADO tailored for extensive games.  Then they propose the SP-PSRO algorithm that enhances APSRO by adding at each iteration a well-chosen mixed strategy.

**Strengths:**

-The new ADO algorithm improves over the DO algorithm by guarantying a monotonically non-increasing exploit ability.

-The two new practical algorithms APSRO and SP-PSRO seem to improve empirically over their direct competitor the PSRO algorithm.

**Weaknesses:**

-While the text is globally easy to read I think that more pointers to the appendix, in particular for the experiments, could improve the clarity of the presented work. Some additional explanations about the computation of $\pi^r$ in the different algorithm is missing at least in the main text.

-The improvement in terms of monotonically of the exploitability curve that motivates the introduction of the different new algorithms is not completely clear in the experiments, especially for the APSRO algorithm vs PSRO algorithm.

**Questions:**

#General comments:

- The SP-PSRO  involves many different components: Q-learning for BR, regret minimization for solving the restricted games, potential reservoir sampling and additional supervised learning to train the mixed policy which results in a rather involved algorithm. Maybe it could be interesting to identify the key components of the SP-PSRO

- It could be useful to describe in details what is the restricted games and how the Nash equilibrium of these games is approximated by regret minimization. And maybe you could also provide more details for the proof of Theorem 3.

- What is the average size of the population in the different experiments? Because one possible drawback of the presented methods is that we have to store all the polices of the populations (and possibly for the average in SP-PSRO). Could you comment on this point?

#Specific comments:

- P3, beginning of Section 3: It is a bit hard to follow this part without knowing in advance what is DO and PSRO.

- P3, (1): It is not completely clear with the current notations if the policy \pi_i \in \Pi_i could be a mixture of the policies in the population or should be exactly one of them. The introduction suggested that the first statement is the correct one? It could be also interesting to remark that a mixture of policy is also a policy somewhere.

- P5 beginning of Section 4: Can you describe precisely how you do regret minimization among the the population of policies in the restricted game? In particular how do you deal with an increasing number of policies?

- P7 bottom: Can you define precisely the time average \bar{\nu}_i. Th whole procedure looks quite involved. i do not really see the motivation behind the averaged best response.

- P8, Section 6.1: It would be interesting to compare also with ADO to see what the new Double oracle brings over PSRO and how well APSRO approximate ADO. Which hyper-parameters did you use and which regret minimizer did you use in APSRO and SP-PSRO? (pointers to the appendix). Why not use the same horizon for all the games? How many seeds did you use in the experiments?

- P8, Section 6.2: Same remark it would be clearer to add reference to the appendix about the details of the experiments. How do you explain intuitively the gap between SP-PSRO and the other algorithms? It seems that that  SP-PSRO converges quickly but then get stuck to a sub-optimal solution, e.g. in Leduc Poker it converges to an exploitability of ~0.4 which is not that good for this game. It is not very clear in Fig 5 that the exploitability of APSRO decrease more monotonically than the one of PSRO contrary to what is claim in the previous sections (in particular for tiny battleship. Could you comment on this point?

- P8, Section 6.2: Same question as above. And would it be possible to use larger horizon in Fig 6 and 5 to see if the exploitabilities of SP-PSRO and APSRO cross or not. How do you compute the average strategy in the experiments?

---

> ### Author Response · Authors · 2023-11-21
> **Author Response**
>
> Firstly, we would like to extend our sincere gratitude for the time and effort you have dedicated to reviewing our paper. We appreciate the opportunity to address the concerns you have raised. We believe that we have sufficiently addressed each of your points and have updated the paper with these changes in blue.
>
> ## Regarding Clarity and Appendix References
> We acknowledge your suggestion for clearer guidance towards the appendix, especially concerning our experimental framework. In our revised manuscript, we have included references to the appendix to clear up confusion. Revised text is in blue.
>
> ## Monotonicity of Exploitability Curve
> APSRO indeed does not show a completely monotone decrease in exploitability due to the imprecision in the best response and no-regret procedure. However, our experiments show that in most settings APSRO does not increase exploitability as much as PSRO. Furthermore, APSRO is a core component of SP-PSRO, which exhibits drastic increases in performance.
>
> ## Complexity of SP-PSRO
> Your observation about the complexity of the SP-PSRO algorithm is well-taken. Appendices F.1 and F.2 demonstrate that each component of SP-PSRO is in fact necessary.
>
> ## Details on Restricted Games and Nash Equilibrium
> In response to your suggestion, we have updated the text with clearer references to the appendix, which contains a detailed description of the restricted games and the methods used for approximating Nash equilibria, ensuring a clearer understanding of these critical aspects. We have also included a proof of Theorem 3.
>
> ## Population Size and Policy Storage
> We have updated the appendix to include data on the population sizes and disk space usage for deep RL experiments. The largest population we generated had 148 policies when running SP-PSRO on 4-Repeated RPS, taking 22MB of storage.
>
> ## Clarifications on Notations and Procedures
> - We have included a reference to DO in the appendix to clear up the beginning of section 3
> - \pi_i \in \Pi_i is indeed a single policy. We have updated the paper to make this clearer.
> - We have included a reference to the exact regret minimization procedure in the appendix at the beginning of section 4
> - We have included a reference to Appendix K.3 for SP-PSRO details of how we compute the time average. We also include in Appendix F.2 ablations showing that the time-average variant of SP-PSRO is necessary. Intuitively, we need to do time-averaging to achieve a good mixed strategy.
> - We did not include ADO in our normal form games because the performance was almost exactly the same as APSRO, so we didn’t want to make it confusing to read. We have included references to the appendix for training details. We didn’t use the same horizon because some games converged faster. Would you like us to include these results for the camera-ready version?
> - We have included a reference to the Appendix for training details. Our intuition, as also described in the paper, is that one should include mixed strategies in the population. Ideally, these mixed strategies will be very close to a NE, so that is why we use a modified fictitious play procedure to produce the mixed strategies. You are right that SP-PSRO seems to have a big boost in performance at the beginning, but we don’t see other methods like PSRO or APSRO overtaking it. This makes sense because we can only improve our population by adding a policy to it, as long as we then output the least-exploitable distribution. As to the non-monotonicity of APSRO in Tiny Battleship, we acknowledge the limitation in predicting when APSRO might outperform PSRO. We hypothesize that APSRO's no-regret procedure was unable to find a satisfactory solution given the small budget allocated to each DO iteration. Hyperparameter tuning may yield better results.  We have, however, observed consistent improvements in most games tested.
> - We do not extend the length of experiments because we do not witness the exploitability lines crossing in normal form experiments, and each run already takes multiple days to complete. We included a reference to the appendix which contains the details of how we compute average strategies.
>
> In conclusion, we have addressed all the concerns you have raised. We are confident that these revisions will significantly enhance the clarity, quality, and impact of our work. Once again, thank you for your constructive feedback and for contributing to the improvement of our paper.

---

### Author Response · Authors · 2023-11-21
**Author General Response**

We have submitted an updated version of our paper, highlighting the revisions in blue text. We have addressed all the comments from the reviewers, except for one from Reviewer C3x7 requesting experiments on larger games, which we could not conduct due to time constraints. We are confident that this revised paper addresses the feedback from the reviewers and hope that reviewers can update their scores accordingly.

---

### Meta-Review · Area_Chair_3t7D · 2023-12-06

**Metareview:**

The paper proposes a new double-oracle algorithm to find an approximate Nash equilibrium in a two-player zero-sum imperfect information game with perfect recall. The approach is new, to the best of my knowledge, and has improvement over the classical double-oracle algorithm with both theoretical (e.g., the monotonic decreasing of exploitability across iterations) and experimental support. The paper is well-written in general, and it reaches a consensus that it is above the acceptance bar. I encourage the authors to incorporate all the feedback in preparing the final version of the paper.

**Justification For Why Not Higher Score:**

Though the proposed approach is new, the paper could have been stronger and make a higher impact if the presentation can be further improved (e.g., more pointers to the appendix to facilitate the reading of the main paper, more careful definitions of notation to improve the rigor of the paper), the experimental results on larger-scale games and ablation studies can be provided, and more clarity on the motivation  can be provided.

**Justification For Why Not Lower Score:**

It is a good paper with novel ideas, new algorithms, and reasonable results.

---

### Decision · Program_Chairs · 2024-01-16

Accept (poster)